# Automating Large-scale *in-silico* Benchmarking for Genomic Foundation Models

## Abstract

The advancements in artificial intelligence in recent years, such as Large Language Models (LLMs), have fueled expectations for breakthroughs in genomic foundation models (GFMs). The code of nature, hidden in diverse genomes since the very beginning of life's evolution, holds immense potential for impacting humans and ecosystems through genome modeling. Recent breakthroughs in GFMs, such as Evo, have attracted significant investment and attention to genomic modeling, as they address long-standing challenges and transform *in-silico* genomic studies into automated, reliable, and efficient paradigms. In the context of this flourishing era of consecutive technological revolutions in genomics, GFM studies face two major challenges: the lack of GFM benchmarking tools and the absence of open-source software for diverse genomics. These challenges hinder the rapid evolution of GFMs and their wide application in tasks such as understanding and synthesizing genomes, problems that have persisted for decades. To address these challenges, we introduce `GFMBench`, a framework dedicated to GFM-oriented benchmarking. `GFMBench` standardizes benchmark suites and automates benchmarking for a wide range of open-source GFMs. It integrates millions of genomic sequences across hundreds of genomic tasks from four large-scale benchmarks, democratizing GFMs for a wide range of *in-silico* genomic applications. Additionally, `GFMBench` is released as open-source software, offering user-friendly interfaces and diverse tutorials, applicable for `AutoBench` and complex tasks like RNA design and structure prediction. To facilitate further advancements in genome modeling, we have launched a public leaderboard showcasing the benchmark performance derived from `AutoBench`. `GFMBench` represents a step toward standardizing GFM benchmarking and democratizing GFM applications.

## 1 Introduction

The central dogma of biology (Crick, 1970) posits that genomes, including DNA and RNA, encode and transmit the genetic information essential for all living systems and underpin the translation of proteins. Despite decades of advancements in molecular biology, deciphering genomes remains a significant challenge (Beaulaurier et al., 2019; Cole et al., 1998; Strous et al., 2006). Researchers have been striving for advanced and efficient genome analysis to better understand and synthesize RNA (Leslie E, 2004) and DNA (Ramadan et al., 2004) genomes. However, the efficiency and performance of conventional bioinformatics approaches (Min et al., 2017; Larranaga et al., 2006) have hardly kept pace with the rapid advancements in high-throughput sequencing technologies (Reuter et al., 2015; Loman et al., 2012). The recent proliferation of foundation models (Akiyama & Sakakibara, 2022; Nguyen et al., 2023; Dalla-Torre et al., 2023) in the natural language processing domain has shown unprecedented potential for modeling complex 'genomic languages' (Nguyen et al., 2023). These models are known as genomic foundation models (GFMs). Such GFMs are so versatile that they not only uncover genomic encoding patterns within DNA and RNA, but also support a diverse array of genomic tasks, such as RNA secondary structure prediction (Seetin & Mathews, 2012), RNA function (e.g., translation efficiency) prediction (Chu et al., 2024), and even RNA molecular design (Westhof et al., 1996; Yesselman et al., 2019; Koodli et al., 2021).

Despite these advancements, the broader adoption of GFMs for genomic research and related fields, including bioscience discovery and therapeutics design, is significantly hindered by the absence of

standardized benchmarks. These benchmarks constitute the foundation for evaluating and comparing model performance, understanding model behavior, building confidence, and promoting the widespread application of GFMs. Unlike the deep learning community in computer vision and natural language processing, where benchmarking has a long-standing tradition, the genomic field faces unique challenges in establishing robust benchmarks due to the following challenges.

- **Data Scarcity and Bias:** A critical challenge is the lack of comprehensive and diverse datasets necessary for robust training and testing of GFMs. In practice, many genomic datasets are limited in scope and size, often exhibiting biases toward specific species or genome sequences. For instance, some GFMs are trained solely on evolutionary conserved sequences (Akiyama & Sakakibara, 2022; Chen et al., 2022; Zhang et al., 2024). This scarcity of diverse datasets significantly hampers the models' ability to generalize and perform effectively across a wide range of genomic contexts. This limitation not only restricts GFM training but also undermines their capacity to discover novel patterns and make accurate predictions in less-studied species.

- **Metric Reliability:** Another major concern affecting model reliability is the inconsistency of metrics used to benchmark performance. Different studies may employ varying metrics or implement the same metrics with minor differences (Post, 2018). This often leads to inconsistent results across studies. For example, Chen et al. (2020) and Fu et al. (2022) has reported significantly different results on the effectiveness of E2EFold (Chen et al., 2020) because of the variations in evaluation metrics. Such inconsistencies can obscure true model performance and hinder the ability to draw reliable conclusions in genomic studies.

- **Reproducibility:** Ensuring reliable reproducibility of GFM experiments across different research environments remains a significant challenge. As reported by (Pineau et al., 2021), differences in computational environments, dataset splits, and even minor code implementation variations can lead to significant discrepancies in results. These inconsistencies hinder the validation and comparison of GFMs. Moreover, the absence of standardized benchmarking practices exacerbates these issues, underscoring the necessity of establishing protocols that can be consistently followed across studies and laboratories to ensure reliable and reproducible research outcomes. In addition, the inherent complexity of GFMs presents formidable roadblocks for domain scientists seeking to identify and implement best practices for GFM building and training tailored to their scientific inquiries. Together, these challenges significantly hamper the democratization of GFMs, limiting their accessibility and adoption across diverse research domains.

- **Adaptive Benchmarking:** As reported in recent studies (Nguyen et al., 2024; Yang & Li, 2024), predictive modeling performance can be significantly enhanced by jointly modeling various genomics, including DNA and RNA. While genomes and proteomes from diverse living systems may share similar patterns in bio-sequence modeling, there is a lack of holistic understanding of GFMs' capabilities beyond their pre-training scenarios. For example, what is the capability of a GFM for structure prediction (Tan et al., 2017; Danaee et al., 2018; Mathews, 2019; Kalvari et al., 2021) when the model was not pre-trained on the structure annotations of the target species? To address this, we developed a novel adaptive benchmarking protocol that enables comprehensive evaluations across a wide range of genomes and species. It is distinguished by its compatibility with diverse GFMs and benchmarks across different modalities of genomic data. Such adaptive benchmarking can facilitate findings from cross-genomic studies and provide valuable insights for future research.

To address these challenges, we develop a dedicated benchmarking toolkit, dubbed `GFMBench`, for GFM-oriented genomics, capable of benchmarking and leveraging GFMs in *in-silico* tasks. This platform champions the following four key characteristics.

- We have collected and integrated 4 large-scale benchmarks, 42 millions of genome sequences from up to 75 genomic datasets, into `GFMBench` to mitigate issues of **data scarcity**. We also perform data filtering for downstream tasks, e.g., structure predictions, that suffer from data leakage, reducing similar sequences and structures. This aids in addressing biases in the learning series and annotated data.
- To mitigate data and implementation bias that break **metric reliability**, we have integrated common metrics and developed automatic performance recording in benchmark evaluations to ensure consistently fair performance.
- To improve the **reproducibility** of benchmark experiments, `GFMBench` exploits the benchmarks compiled in a unified protocol, specifying detailed metadata and benchmark settings that follow the FAIR principles (Wilkinson et al., 2016). e.g., hyperparameters and dataset splits, that

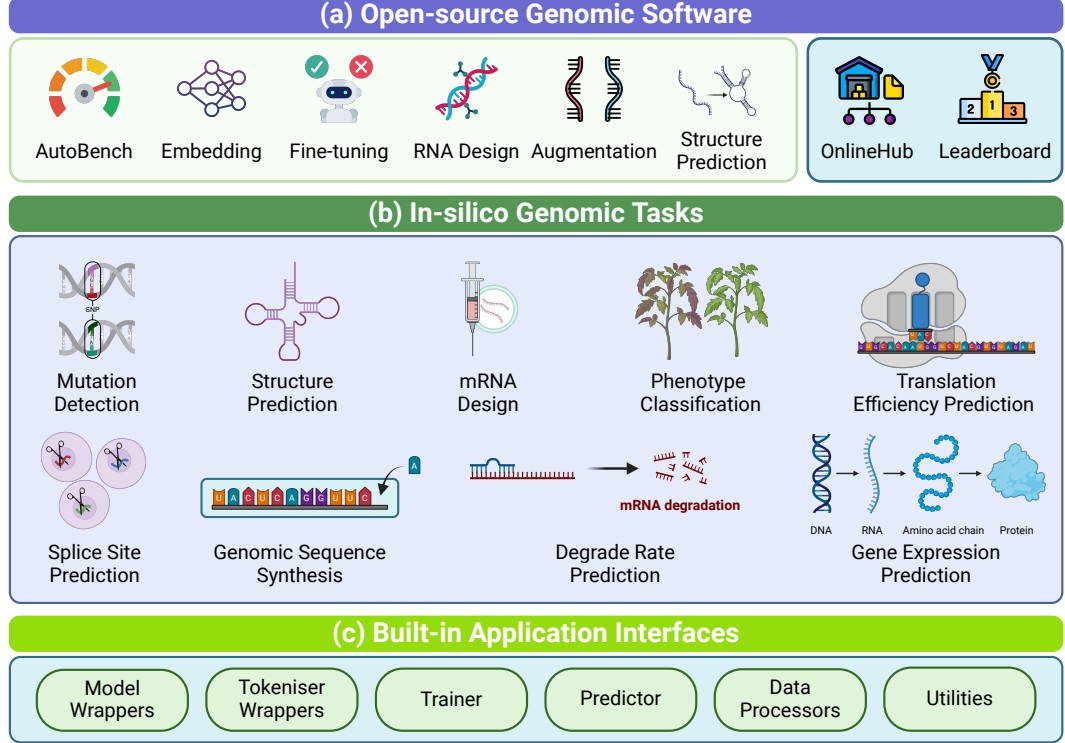

Figure 1: (a) shows the available tools in the open-source software, including the `AutoBench` pipeline and genome embedding extraction. `GFMBench` also launches an online hub and leaderboard to support GFM development. (b) illustrates the diverse genomic tasks supported by `GFMBench`, enabling both benchmarking and fine-tuning. This allows even novices in GFM to implement and fine-tune models without writing custom code. `GFMBench` includes common task templates and offers built-in interfaces for implementing new tasks. In addition to the fine-tuning interfaces, `GFMBench` provides user-friendly tools for running inferences and deployments.

    may lead to performance variance across models and datasets. On the other hand, the stark lack of biologist-friendly GFM-oriented genomics software like ViennaRNA[1], necessitates expertise in language modeling and bioinformatics to democratize GFMs and hinders the exploration of leveraging GFMs in genome modeling. The absence of standardized solutions may exacerbate conclusion inconsistencies in genome modeling studies, as custom-built solutions may introduce uncertainties in model reliability.

- **Adaptive benchmarking** is supported in `GFMBench`. It is distinguished by its compatibility with diverse GFMs and benchmarks across different modalities of genomic data. Such adaptive benchmarking can facilitate findings from cross-genomic studies and provide valuable insights for future research. For example, Yang & Li (2024) showed that RNA structure pre-training significantly improves model performance on DNA genomic benchmarks, indicating that structural information is vital even for DNA genomes.

## 2 PROPOSED GFMBENCH

`GFMBench` is an open-source benchmarking software platform for GFMs, and its architecture is shown in Figure 1. Figure 1 (a) presents the available toolkit for genomics, including the `AutoBench` pipeline and genome embedding extraction, among other features. Figure 1 (b) shows the diverse set of genomic tasks supported by `GFMBench` for both benchmarking and fine-tuning. Moreover, Figure 1 (c) illustrates the user-friendly built-in interfaces in `GFMBench` for implementing and fine-tuning new models, as well as for running inferences and deployments. We delineate the components in the following sections.

---

[1]`https://www.tbi.univie.ac.at/RNA`

`GFMBench` is an open-source benchmarking software platform for GFMs, and its architecture is shown in Figure 1 Figure 1. Figure 1 (a) presents the available tools for genomics, including the `AutoBench` pipeline and genome embedding extraction. Figure 1 (b) displays the genomic tasks that `GFMBench` supports for benchmarking and fine-tuning. Figure 1 (c) shows the user-friendly interfaces for implementing new models, as well as for running inferences and deployments. We describe the components in detail below.

## 2.1 AUTOBENCH PIPELINE

`AutoBench` is an automated benchmarking solution for genomics, involving the concepts of benchmark suite standardization, open-source GFM compatibility, and metric implementation. In `AutoBench`, four standardized benchmark suites can be evaluated with existing open-source GFMs, alleviating the data scarcity problem. Public and custom metric implementations are also supported, allowing users to benchmark models against specific performance requirements. The metrics in `AutoBench` are recommended and distributed as part of the benchmark suites to ensure metric reliability for GFMs. It prioritizes the standardization of benchmark suites and protocols to minimize benchmarking uncertainties.

**Benchmark Suites** It has been recognized that comprehensive benchmark suites are crucial for language modeling evaluation. To be more specific, genome languages are complicated and far from manipulation like natural languages, because the understanding of genetic information is challenging for GFMs. For example, single nucleotide variants (SNVs) (Miladi et al., 2020) and single nucleotide polymorphisms (SNPs) (Rafalski, 2002) often lead to significant shifts in phenotypes, while the critical base differences in such long sequences are sparse (Shastry, 2002) and difficult for GFMs to perceive. Moreover, each genome of a species can essentially be regarded as a different language, which further complicates genome modeling. To achieve robust evaluations, `GFMBench` has integrated four large-scale benchmarks that enable broad evaluations of *in-silico* tasks. These benchmarks comprise 42 million genomic sequences and help alleviate the problem of **data scarcity and bias**, allowing for precise and generalizable performance assessments of GFMs.

The brief introductions of the available benchmarks for `AutoBench` in `GFMBench` are as follows:

- **RNA Genomic Benchmark (RGB)** (Yang & Li, 2024). RGB consists of 7 challenging single-nucleotide (SN) level genome understanding tasks, curated or collected from published sources, as shown in Table 5. It aims to benchmark GFMs in SN-level modeling tasks such as predicting mRNA degradation rates and secondary structures. The sequences in RGB range from 107 to 512 bases, making them suitable for verifying RNA model efficacy. These downstream tasks in RGB assemble the first comprehensive RNA benchmark to assess the multi-species and SN-level modeling capabilities of GFMs. For detailed information on each dataset, such as their sources and sizes, please refer to Appendix B.1.
- **Plant Genomic Benchmark (PGB)**. PGB[2] (Mendoza-Revilla et al., 2023) shown in Table 7, provides a large-scale DNA benchmark suite designed to evaluate GFMs specialized in plant biology. PGB involves 8 types of DNA downstream subtasks, including a range of critical tasks such as promoter strength prediction and gene expression regression. There are 28 datasets in total, with millions of DNA sequences in PGB, and the sequence lengths are up to 6,000, which is quite long for most genomic FMs. Since the original evaluation protocol is not publicly available, we have re-implemented the auto-benchmark for all the subtasks from PGB in `GFMBench`.
- **Genomic Understanding Evaluation (GUE)** (Zhou et al., 2023). GUE is a multi-species genome classification benchmark that includes 36 datasets across 9 important genome analysis tasks, with input lengths ranging from 70 to 10,000. The benchmark covers a variety of species, including humans, fungi, viruses, and yeast, and explicitly defines evaluation metrics for each task, ensuring fair comparisons across different models. GUE consists of 7 genome sequence classification problems and 28 datasets, focusing on sequences with input lengths up to 1,000, offering a robust testing ground for models handling longer genomic sequences.
- **Genomic Benchmarks (GB)** (Grešová et al., 2023). GB is an early collection of DNA genome datasets aimed at evaluating the genomic sequence classification performance of deep learning models. The benchmark includes 9 datasets focusing on various regulatory elements, such as promoters, enhancers, and open chromatin regions, across different species like humans, mice, and

---

[2]https://huggingface.co/datasets/InstaDeepAI/plant-genomic-benchmark

roundworms. The downstream datasets aim to standardize comparisons, promote reproducibility, and drive innovation in genomic modeling.

**Benchmark Standardization**   The universal protocols for benchmark suites benchmarks has been absent in existing works, leading to performance biases caused by in-consistent implementations, such as hyperparameters. To tackle this problem, we propose the benchmark standardization, compiling suites with comprehensive components that guarantee identical benchmarking results, such as **metadata**, **hyperparameter settings**, **custom code implementations**, **metrics specifications**.

Benchmark **metadata** is essential for ensuring adherence to the FAIR principles (Wilkinson et al., 2016). We included primary keys in the metadata, including data sources, species information, genome specifications (e.g., DNA, RNA), and data scale measures. We also encourage to include custom keys in the future benchmarks to allow users to interpret such benchmark suites in depth. To increase benchamrking reproducibility, We freeze the **hyperparameter settings** in the standardized benchmark suites, because a slight change of hyperparameter may lead to significant variances (Post, 2018), such as batch sizes and optimizers. We also notice that the some genomic tasks requires **custom codes** to complete the benchmark, e.g., codes to process the data and implement the model and tokenizers. Therefore, GFMBench can parse custom codes in the configuration corresponding to each task and override builtin behaviors. In GFMBench, we mirrored a diverse set of common metrics from scikit-learn[3] to supported diverse tasks. However, we are aware of some tasks that demand special unsupported metrics, GFMBench will load the load **metric** implementations in the standardized suites like the custom codes. In GFMBench, we have implemented a diverse set of common performance metrics, such as F1 score, MCC and AUC. Apart from the built-in metrics, custom metrics can be included in benchmark suites to eliminate the performance variance caused by implementations.

The precompiled benchmark suites are distributed and evaluated according to GFMBench. This standardization not only enhances the **reproducibility** of benchmark results but also facilitates a more nuanced understanding of model strengths and limitations across diverse genomic tasks.

**Genomic Foundation Models**   The primary challenges stem from the heterogeneity of GFM architectures such as Transformers (Vaswani et al., 2017; Lin et al., 2023), Hyena (Poli et al., 2023; Nguyen et al., 2023; 2024) and Mamba (Gu & Dao, 2023; Schiff et al., 2024b). These versatile implementations of GFMs require distinct environment and package requirements, as we as different interfaces for initialization, training and inference, leading to inefficient GFMs performance evaluation across benchmarks. Moreover, there have been attempts of genome tokenization methods (Zhou et al., 2023; Nguyen et al., 2023; Dalla-Torre et al., 2023; Li et al., 2024). The tokenization of genome sequences encountered significant variances between different downstream tasks, such as k-mers (Yang et al., 2023; Dalla-Torre et al., 2023), Byte Pair Encoding (BPE) (Devlin et al., 2019; Zhou et al., 2023), and Single Nucleotide Tokenizers (SNT) (Nguyen et al., 2023; Chen et al., 2023; Yang & Li, 2024). These tokenization methods feature different implementations and are tailored to be compatible with particular GFMs. In instances where tokenizers are incorrectly instantiated or utilized, this can lead to reports of unreliable performance metrics.

To standardize benchmarking across diverse GFMs with respect to specialized tokenizers, we have developed wrapper templates to unify interfaces and tokenizers, streamlining elastic benchmarking of open-source or customized GFMs. For example, GFMBench is capable of accommodating GFMs integrating RNA secondary structures modeling (Yang & Li, 2024) with a simple model wrapper while existing benchmark tools have yet to achieve such flexibility. We have supported an array of open-source GFMs in GFMBench, detailed in Appendix C.3, and tutorials for adapting future GFMs will be released along with the open-source repository.

**AutoBench for Adaptive Benchmarking**   AutoBench parses the configuration[4] of standardized benchmark suites, automates the evaluation processes via unified interfaces, **adaptive benchmarking** benchmarking among diverse GFMs and suites is seamless in GFMBench, i.e., without any modification of the command. The compiled suites offer standard benchmark

---

[3] `https://scikit-learn.org`

[4] We show an example configuration of RGB at `https://tinyurl.com/GFMBench-Demo/examples/RGB/RNA-mRNA/config.py`

processes among various GFMs and versatile data modalities from diverse species. We provide a tutorial for `AutoBench` in `https://tinyurl.com/GFMBench-Demo/examples/AutoBench_Tutorial.ipynb`. The rationale of adaptive benchmarking in genomics is multifaceted. Firstly, given the vast diversity of genomic data across species, adaptive benchmarking can reveal a GFM's potential for cross-species applications, a critical factor in comparative genomics and evolutionary studies. Finally, GFMs may exhibit surprising proficiency in tasks they weren't explicitly trained for, potentially uncovering novel applications or insights into the relationship between different genomic tasks. Secondly, it allows researchers to gauge models' ability to capture genomic knowledge rather than task-specific patterns, and it stress-tests GFMs under scenarios of underrepresented genomic sequences or tasks. This helps identify limitations or biases in the models that may not be apparent in their primary training domains. Finally, the adaptive benchmarking calls for standardized and universal platforms that accelerate the evolution of GFMs with largely decreased requirements of expertise on benchmarking.

## 2.2 GENOMICS SOFTWARE

GFMs have been extensively used for proof-of-concept *in-silico* experiments, such as secondary structure and translation efficiency predictions, but have yet to be widely acknowledged in *in-vivo* scenarios. Existing GFM studies generally require significant expertise in both NLP and molecular biology, which hampers the widespread adoption of GFM-guided genome analysis. Therefore, `GFMBench` has been designed as open-source software dedicated to genome modeling, similar to ViennaRNA.

**Genomic Toolkit** As open-source software for universal genomics, we have curated a range of features, including genome embedding extraction, genome data augmentation, and common genomic tasks such as RNA design. Additionally, users can utilize the user-friendly application interfaces (APIs) to complete genomic downstream tasks without any prior knowledge, such as training and testing a GFM for RNA sequence classification, enabling users to easily integrate `GFMBench` into their workflows. We provide some code examples[5] to demonstrate the toolkit's usage, e.g., automated benchmarking, showcasing the utmost utility of `GFMBench` in genomic modeling.

**Online Hub** Inspired by successful practices within the NLP community, we have developed an online hub designed to host and distribute a wide range of resources. This hub provides a centralized platform where users can easily access precompiled benchmark suites and fine-tuned models, enabling efficient testing and experimentation for GFMs. It simplifies the workflow for both novices and experts by offering ready-to-use benchmarks and models that can be downloaded or directly integrated into their research pipelines. This hub is community-driven, encouraging collaboration and innovation by allowing researchers from around the world to contribute their own benchmarks, models, and evaluation metrics.

**Leaderboard** We have developed an open GFM leaderboard, showcasing detailed task-wise performance for DNA and RNA downstream tasks. Please refer to the current leaderboard in Appendix D. To mitigate benchmarking bias, performance fidelity and comparison fairness are ensured as the evaluation results are generated by `AutoBench`, eliminating manual interventions. The leaderboard helps researchers identify GFMs' strengths and weaknesses on a task-by-task basis, aiding them in selecting the most appropriate models for their tasks. The leaderboard also accepts community contributions, such as new benchmark result submissions.

**Design Principles** To improve the sustainability of `GFMBench`, it has been specifically engineered to simplify the complexities associated with genomic modeling, ensuring research reliability and enhancing prediction efficiency, adhering to the following design principles:

- **Utility:** The software provides user-friendly interfaces for hundreds of downstream tasks, such as RNA design. It allows the training and inference of genomic tasks to be conducted with minimal coding.
- **Simplicity:** Comprehensive tutorials are available to assist GFM novices, covering a range of topics from data curation and augmentation to GFM fine-tuning and inference.

---

[5]`https://tinyurl.com/GFMBench-Demo/examples/readme.md`

- **Diversity:** A broad spectrum of benchmarks and GFMs is included to support the development of various task types and model architectures, fostering innovation and experimentation in genomic research.
- **Extensibility:** The software supports the benchmarking of custom tokenizers, GFMs, and metrics without the need to modify existing source code, offering flexibility and adaptability to meet specific research needs.
- **Community:** An interactive leaderboard showcases detailed performance evaluations of GFMs, promoting transparency and competitive development within the field. Additionally, a container-ized evaluation environment is provided to minimize the impact of software variability on the reliability of benchmarking.

## 3 BENCHMARK RESULTS

In this section, we present a comprehensive evaluation of GFMs using the `GFMBench`. We report the performance of open-source GFMs across four major genomic benchmark suites, i.e., RGB, PGB, GUE and GB, highlighting GFMs' strengths and weaknesses across diverse genomic tasks and datasets. To mitigate potential class imbalance issues, we adopt the macro F1 score as the metric in classification tasks, replacing accuracy where necessary.

### 3.1 RNA GENOMIC BENCHMARK (RGB)

The RGB comprises seven challenging single-nucleotide level RNA modeling tasks, designed to evaluate models' fine-grained capabilities in understanding RNA sequences, such as predicting RNA structures. These tasks include mRNA degradation rate prediction, single-nucleotide modification detection (SNMD), single-nucleotide modification regression (SNMR), and RNA secondary structure prediction tasks such as Archive2, Stralign, and bpRNA. Additionally, EternaV2 evaluates models on RNA design tasks.

Table 1 presents the performance of various GFMs on the RGB tasks. Overall, OmniGenome achieves the best performance across all tasks, highlighting its exceptional capability in RNA structure modeling. This superior performance can be attributed to OmniGenome's integration of structural information into its modeling process, which is particularly beneficial for tasks requiring secondary structure prediction.

Table 1: The performance of `GFMBench` and baseline models on the RGB, with results averaged based on five random seeds. "N.A." means not available for predictive tasks.

| Model | mRNA | SNMD | SNMR | Archive2 | Stralign | bpRNA | EternaV2 |
|---|---|---|---|---|---|---|---|
| | RMSE | AUC | F1 | F1 | F1 | F1 | Accuracy |
| ViennaRNA | N.A. | N.A. | N.A. | 73.99 | 74.09 | 65.03 | 33 |
| MXFold2 | N.A. | N.A. | N.A. | 90.09 | 97.01 | 64.99 | N.A. |
| Ufold | N.A. | N.A. | N.A. | 89.78 | 95.76 | 78.38 | N.A. |
| DNABERT2 | 0.8158 | 49.94 | 15.86 | 55.73 | 64.09 | 33.77 | 0 |
| HyenaDNA | 0.8056 | 53.32 | 39.80 | 71.18 | 91.24 | 57.43 | 0 |
| Caduceus | 0.8026 | 57.01 | 39.59 | 74.37 | 92.28 | 59.76 | 0 |
| NT-V2 | 0.7826 | 50.49 | 26.01 | 68.36 | 83.18 | 56.95 | 0 |
| Agro-NT | 0.7830 | 49.99 | 26.38 | 62.81 | 72.54 | 46.87 | 0 |
| SpliceBERT | 0.7340 | 58.11 | 46.44 | 79.89 | 93.81 | 71.59 | 3 |
| 3UTRBERT | 0.7772 | 50.02 | 24.01 | 68.62 | 88.55 | 57.90 | 0 |
| RNABERT | 0.8087 | 51.32 | 29.14 | 24.66 | 83.68 | 47.96 | 0 |
| RNA-MSM | 0.7321 | 57.86 | 45.22 | 68.72 | 91.15 | 64.44 | 2 |
| RNA-FM | 0.7297 | 59.02 | 42.21 | 82.55 | 95.07 | 78.16 | 4 |
| OmniGenome | **0.7121** | **64.13** | **52.44** | **91.89** | **98.21** | **83.18** | 84 |

In particular, OmniGenome significantly outperforms other models on the mRNA degradation rate prediction task, achieving an RMSE of 0.7121, compared to the second-best RMSE of 0.7297 by RNA-FM. Similarly, for the SNMD task, OmniGenome achieves an AUC of 64.13, surpassing the second-best score of 59.02 by RNA-FM. These results indicate that OmniGenome effectively captures single-nucleotide level variations, which are crucial in RNA function and regulation. Furthermore, in the secondary structure prediction tasks (Archive2, Stralign, bpRNA), OmniGenome demonstrates superior performance, highlighting its proficiency in modeling RNA secondary struc-

Table 2: Performance of open-source GFMs on PGB, where the results are re-implemented based on our evaluation protocol. "PolyA" stands for Polyadenylation, "Chrom Acc" for Chromatin Accessibility, "Prom Str" for Promoter Strength, "Term Str" for Terminator Strength, "Splice" for Splice Site, "Gene Exp" for Gene Expression, and "Enh Reg" for Enhancer Region.

| Model | PolyA | LncRNA | Chrom Acc | Prom Str | Term Str | Splice | Gene Exp | Enhancer |
|---|---|---|---|---|---|---|---|---|
| | F1 | F1 | F1 | RMSE | RMSE | F1 | RMSE | F1 |
| DNABERT2 | 41.35 | 72.55 | 61.49 | 0.99 | 0.24 | 45.34 | 14.78 | 36.40 |
| HyenaDNA | 83.11 | 58.21 | 52.20 | 0.88 | 0.26 | 90.28 | 14.79 | 66.17 |
| Caduceus | 70.89 | 68.40 | 64.53 | 0.91 | 0.26 | 78.51 | 14.72 | 60.83 |
| NT-V2 | 71.26 | 73.08 | 65.71 | 0.81 | 0.27 | 95.05 | 14.79 | 73.89 |
| Agro-NT | 78.89 | 67.24 | 63.27 | 0.94 | 0.78 | 88.45 | 15.56 | 62.83 |
| SpliceBERT | 65.23 | 71.88 | 63.62 | 0.75 | 0.22 | 96.45 | **14.70** | 69.71 |
| 3UTRBERT | 76.48 | 70.75 | 63.71 | 1.04 | 0.36 | 94.44 | 14.87 | 71.67 |
| RNA-BERT | 78.54 | 61.99 | 48.94 | 1.81 | 0.38 | 94.45 | 14.89 | 57.61 |
| RNA-MSM | 84.25 | 67.49 | 53.52 | 1.28 | 0.28 | 95.49 | 14.87 | 61.45 |
| RNA-FM | 84.94 | 68.75 | 54.92 | 0.95 | 0.27 | 95.95 | 14.83 | 57.14 |
| OmniGenome | **87.55** | **77.96** | **67.69** | **0.59** | **0.18** | **98.41** | 14.71 | **79.77** |

tures. This can be attributed to OmniGenome's incorporation of structural context during pretraining, which enhances its ability to understand and predict RNA folding patterns. One limitation observed is that models not specifically designed for RNA tasks, such as DNABERT2 and HyenaDNA, perform poorly on RNA-specific tasks. This underscores the importance of tailoring GFMs to the specific characteristics of RNA sequences.

In summary, the RGB results highlight the critical role of structural modeling in RNA genomics and demonstrate the effectiveness of OmniGenome in capturing complex RNA features. Future GFMs may benefit from incorporating structural information to enhance performance on RNA-related tasks.

## 3.2 PLANT GENOMIC BENCHMARK (PGB)

The PGB comprises DNA-based tasks focused on plant biology, and the detailed task descriptions can be found in the Appendix B. The sequences in PGB contain up to 6,000 bases, presenting challenges for models in handling long genomic sequences. Table 2 summarizes the performance of various GFMs on the PGB tasks. Resembling the results of RGB, OmniGenome achieves top-tier performance across most tasks, even though it was only trained on RNA. This suggests that OmniGenome generalizes well to DNA-based tasks, likely due to shared sequence motifs and structural similarities between RNA and DNA.

In the PolyA task, OmniGenome achieves an F1 score of 87.55, outperforming the second-best model, RNA-FM, which achieves 84.94. Similarly, for the LncRNA task, OmniGenome attains an F1 score of 77.96, significantly higher than the second-best score of 73.08 by NT-V2. OmniGenome excels in the Splice Site prediction task, achieving an F1 score of 98.41, surpassing the second-best score of 96.45 by SpliceBERT. This suggests that OmniGenome effectively captures sequence motifs important for splicing, which is crucial in gene expression regulation. These results indicate that GFMs incorporating structural context, like OmniGenome, can generalize effectively across different genomic modalities (RNA and DNA) and species (plants). The strong performance of OmniGenome on DNA-based tasks suggests that structural modeling enhances the understanding of genomic sequences beyond the specific type of nucleic acid. However, it's also observed that some models specifically designed for DNA tasks, such as NT-V2 and SpliceBERT, perform competitively on certain tasks. This underscores the importance of task-specific pretraining and the potential benefits of integrating both sequence and structural information in GFMs.

In summary, the PGB results highlight the potential for cross-modal generalization in GFMs and the value of incorporating structural context to enhance performance on diverse genomic tasks.

## 3.3 GENOMIC UNDERSTANDING EVALUATION (GUE)

The GUE is a multi-species benchmark like RGB and PGB, but focuses on the non-plant genomes. The sequences in GUE range in length and complexity, providing a robust assessment of GFMs' abilities to generalize across species and genomic tasks. Table 3 presents the performance of various

GFMs on the GUE tasks. While OmniGenome does not achieve the highest performance on all tasks, it consistently delivers competitive results, demonstrating strong cross-species generalization despite being primarily trained on RNA data.

Table 3: Performance of open-source GFMs on GUE, where the results are re-implemented based on our evaluation protocol. The performance for each task is the average macro F1 score in all sub-datasets.

| Model | Yeast EMP | Mouse TF-M | Virus CVC | Human TF-H | Human PD | Human CPD | Human SSP |
|---|---|---|---|---|---|---|---|
| | F1 | F1 | F1 | F1 | F1 | F1 | F1 |
| DNABERT-2 | 75.85 | **86.23** | 58.23 | 81.80 | 90.17 | 82.57 | 85.21 |
| HyenaDNA | 73.08 | 73.44 | 27.59 | 77.62 | 91.19 | 84.31 | 83.34 |
| Caduceus | 73.49 | 78.18 | 27.49 | 79.56 | 89.13 | 85.09 | 81.82 |
| NT-V2 | 74.93 | 78.10 | 32.71 | 79.12 | 90.87 | 84.70 | 84.13 |
| SpliceBERT | 77.66 | 84.97 | 47.17 | **82.77** | **92.24** | 83.96 | **93.81** |
| 3UTRBERT | 71.89 | 71.46 | 34.84 | 74.85 | 82.37 | **90.51** | 81.95 |
| RNA-BERT | 60.14 | 59.83 | 21.08 | 67.48 | 79.87 | 76.25 | 44.75 |
| RNA-MSM | 64.99 | 79.15 | 51.81 | 78.72 | 91.28 | 85.42 | 84.24 |
| RNA-FM | 74.41 | 78.24 | 52.22 | 79.27 | 92.18 | 86.05 | 84.76 |
| OmniGenome | **78.51** | 84.72 | **64.41** | 81.73 | 90.04 | 85.22 | 90.39 |

In the Yeast EMP task, OmniGenome achieves the highest F1 score of 78.51, slightly outperforming SpliceBERT of 77.66. For the Virus CVC task, OmniGenome also achieves the best performance with an F1 score of 74.72, indicating its strong ability to model viral genomic sequences. However, for tasks like Human TF-H and Human SSP, models like SpliceBERT and DNABERT2 achieve higher scores. This suggests that these models may be better optimized for human genomic sequences or specific tasks like splice site prediction. The results on GUE highlight the challenges in developing GFMs that generalize across different species and genomic tasks. While OmniGenome demonstrates strong cross-species performance, there is variability depending on the specific task and species. These findings suggest that combining the strengths of different GFMs or developing ensemble methods could be a fruitful direction for future research. Additionally, incorporating more diverse training data and task-specific fine-tuning may enhance the performance of GFMs across a broader range of tasks.

## 3.4 GENOMIC BENCHMARKS (GB)

GB is a collection of DNA genome datasets aimed at evaluating the performance of models on sequence classification tasks involving regulatory elements such as promoters, enhancers, and open chromatin regions across different species including humans, mice, and roundworms. Table 4 shows the performance of various GFMs. The tasks are denoted by their species and regulatory elements, and the acronyms are explained in Appendix B.4.

Table 4: Performance of open-source GFMs on GB, where the results are re-implemented based on our evaluation protocol.. The performance (macro F1) for each task is the average macro F1 score across all sub-datasets.

| Model | DEM | DOW | DRE | DME | HCE | HEE | HRE | HNP | HOR |
|---|---|---|---|---|---|---|---|---|---|
| | F1 | F1 | F1 | F1 | F1 | F1 | F1 | F1 | F1 |
| DNABERT-2 | 92.67 | 95.17 | 43.77 | 77.21 | **75.58** | 80.66 | 78.14 | 85.80 | 68.03 |
| HyenaDNA | 88.21 | 94.13 | 70.11 | 76.44 | 70.38 | 79.58 | 96.33 | 85.99 | 67.03 |
| Caduceus | 92.13 | 94.74 | 72.03 | 75.61 | 70.20 | 76.47 | 79.16 | 84.36 | 63.17 |
| NT-V2 | 91.66 | 94.32 | **78.20** | **81.72** | 71.98 | 79.85 | 93.30 | 85.30 | 68.53 |
| SpliceBERT | **94.72** | **96.42** | 72.29 | 74.70 | 73.50 | 79.60 | 95.23 | **89.57** | 68.89 |
| 3UTRBERT | 89.50 | 90.22 | 74.35 | 80.14 | 70.23 | 76.33 | **98.47** | 82.49 | 66.78 |
| RNA-BERT | 76.56 | 62.17 | 50.11 | 60.79 | 66.69 | 63.29 | 46.57 | 73.80 | 56.59 |
| RNA-MSM | 79.38 | 93.71 | 54.13 | 75.90 | 69.79 | 78.07 | 94.87 | 84.28 | 63.93 |
| RNA-FM | 91.53 | 95.49 | 74.77 | 79.74 | 71.62 | 80.03 | 95.72 | 87.14 | **69.38** |
| GFMBench | 94.16 | 93.49 | 77.17 | 80.34 | 73.51 | **82.23** | 95.66 | 87.87 | 68.97 |

In the DEM and DOW tasks, SpliceBERT achieves the highest F1 scores, with OmniGenome closely following in DEM and RNA-FM in DOW. For the DRE task, NT-V2 achieves the best performance with an F1 score of 78.20, with OmniGenome performing closely. In the HEE task, OmniGenome attains the highest F1 score of 82.23, surpassing the second-best score of 80.03 by RNA-FM. This indicates OmniGenome's effectiveness in modeling human enhancer regions. From a global perspective, these results demonstrate that while different GFMs excel in specific tasks, OmniGenome

consistently performs well across various genomic benchmarks, highlighting its versatility. The performance variations across models suggest that task-specific features and training data significantly impact model efficacy. A limitation observed is that GFMs primarily trained on RNA data, like RNA-BERT and RNA-MSM, lost on DNA-based tasks. This underscores the importance of training data relevance and the potential need for multimodal pretraining strategies.

In conclusion, the GB results emphasize the need for GFMs that can generalize across different genomic tasks and species. Integrating structural information, as done in OmniGenome, appears to enhance model performance on complex genomic tasks.

### 3.5 OVERALL DISCUSSION

Our comprehensive evaluation across four genomic benchmarks reveals that OmniGenome consistently achieves top-tier performance, particularly excelling in tasks that involve structural modeling of RNA sequences. The integration of structural information in OmniGenome enhances its ability to capture complex sequence features, which is advantageous across diverse genomic tasks. While OmniGenome demonstrates strong performance even on DNA-based tasks, models specifically tailored to certain tasks or species, such as SpliceBERT and DNABERT2, sometimes outperform OmniGenome in those specific contexts. This suggests that task-specific or species-specific pretraining can provide benefits, and there is potential for combining the strengths of different models.

The absence of results for certain models on some benchmarks (e.g., RNA-BERT, RNA-MSM, and RNA-FM on GUE) highlights the challenges in benchmarking GFMs across diverse datasets. Differences in model architectures, pretraining data, and tokenization strategies can impact a model's applicability to specific tasks. Future work should focus on developing unified evaluation protocols and improving the interoperability of GFMs. An important consideration is the need for detailed descriptions of the models evaluated, including their architectures, pretraining data, and key features. This information is crucial for understanding the factors contributing to their performance and for reproducing results. We acknowledge that such details are essential and included in Appendix B.

Overall, our comprehensive benchmarking highlights the importance of integrating structural information into GFMs and suggests that models capable of capturing both sequence and structural features offer improved performance across a range of genomic tasks. This work provides valuable insights for the development of next-generation GFMs and underscores the need for continued efforts in benchmarking to drive advancements in genomic modeling.

## 4 RELATED WORKS

The GFM-oriented platforms, such as benchmarking and application toolkits, have been investigated but have yet to be revolutionised. For example, there are several benchmarking studies such as RNABench (Runge et al., 2024), GenBench (Liu et al., 2024), BEACON (Ren et al., 2024), and DEGB (West-Roberts et al., 2024), and the application software like Kipoi[6] (Avsec et al., 2019). Overall, these benchmarking tools do not prioritise the standardisation and automation of GFM benchmarking and generally focus on specific scenarios such as DNA benchmarking. On the other hand, there is no GFM-dedicated software which leverages the unprecedented capability of GFMs in the wide applications of *in-silico* genomics. Please find more details of the related works in Appendix A.

## 5 CONCLUSION

We propose `GFMBench` in this work to address the challenges of GFMs in benchmarking and application. `GFMBench` tackles the crux lies in the evaluation of modeling the complex 'genomic language' of DNA and RNA by integrating four large-scale benchmarks and 42 million genome sequences from 75 datasets, to support the evaluation of 10+ open-source GFMs. Moreover, `GFMBench` is an open-source software that simplifies the pipelines of genomic modeling, ensuring *in-silico* genomic research reliability and efficiency, and promoting community collaboration through a dedicated leaderboard.

---

[6] https://kipoi.org

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

## A    RELATED WORKS

### A.1    BENCHMARK

Recognizing the critical role of benchmarking in genomic modeling, several tools have been developed to evaluate genomic models. Among these are RNABench (Runge et al., 2024), GenBench (Liu et al., 2024), BEACON (Ren et al., 2024), and DEGB (West-Roberts et al., 2024).

RNABench focuses on a set of benchmarks, such as RNA secondary structure prediction, and lacks support for evaluating the latest pre-trained models. GenBench is a modular DNA benchmarking framework that provides a DNA evaluation solution but does not extend to RNA benchmarking, and it may not prioritize user-friendliness. BEACON is a recent benchmarking tool aimed at RNA foundational models, offering some RNA evaluation datasets. However, it may lack benchmarking scalability and the complexity of its environment setup poses challenges for novices. DEGB serves as an evaluation benchmark for genomic embeddings, supporting both amino acids and nucleic acids. Its main limitation lies in the small scale of its evaluation benchmarks, and it does not support downstream applications of GFMs. Classic genomic modeling tools like Kipoi[7] (Avsec et al., 2019) have been developed to standardize access to trained models for genomic sequence analysis, offering

---

[7]https://kipoi.org

a repository of models. However, Kipoi focuses on providing access to classic models, not GFMs, rather than benchmarking comprehensively.

There are some protein benchmarking tools, such as ProteinGym (Notin et al., 2024), Flip (Dallago et al., 2021) and Peer (Xu et al., 2022), to name a few. ProteinGym is a large-scale benchmarking tool focused on protein fitness prediction and design. It provides over 250 deep mutational scanning assays, offering a standardized dataset to evaluate machine learning models across millions of mutated protein sequences. ProteinGym is designed to assess both zero-shot and supervised models, particularly in predicting the effects of mutations and aiding protein engineering for applications like genetic disease, agriculture, and healthcare. Flip provides a benchmark for predicting the protein sequence-function relationship, a critical aspect of protein engineering. It includes data for tasks such as adeno-associated virus stability, protein domain stability, and thermostability from multiple protein families. Flip is designed to evaluate model generalization under various conditions, such as low-resource or extrapolative scenarios. Its datasets are curated to assess the capacity of models to predict functional properties of proteins in real-world protein engineering tasks. Peer is a comprehensive multi-task benchmark that offers 17 tasks across five categories, including protein function prediction, localization, structure, and interaction predictions. It evaluates a wide range of machine learning methods, from traditional approaches to large pre-trained protein language models. Peers' broad scope helps assess model performance in different protein-related tasks, contributing to advancements in protein sequence understanding and engineering.

Existing tools do not adequately address the challenges of comprehensive, large-scale evaluation of RNA and DNA GFMs. They often lack support for downstream applications and do not facilitate the ease of use or scalability necessary to catalyses the democratization and revolution of GFM research. This gap has motivated the development of a new benchmarking tool designed to cover a broad spectrum of foundational DNA and RNA models and provide an extensive benchmarking suite.

## A.2 GENOMIC FOUNDATION MODELS

In recent years, the modeling of biological sequences, including DNA, RNA, and proteins, has garnered significant attention. Protein modeling, exemplified by works such as AlphaFold (Jumper et al., 2021; Evans et al., 2021; Abramson et al., 2024) and ESM (Lin et al., 2022), has advanced considerably over the past years, outpacing developments in DNA and RNA modeling.

In the domain of genomic sequence modeling, early efforts focused on adapting natural language processing architectures to handle genomic data. For instance, DNABERT (Ji et al., 2021) repurposed the BERT (Devlin et al., 2019) architecture for genomic sequences, demonstrating preliminary success on *in-silico* genomic tasks. Building upon this, DNABERT2 (Zhou et al., 2023) introduced improvements by replacing k-mer tokenization with byte-pair encoding (BPE) tokenization, enhancing model performance across multiple species.

To explore the capabilities of large-scale foundation models (FMs), the Nucleotide Transformers V2 (Dalla-Torre et al., 2023), AgroNT (Mendoza-Revilla et al., 2023), and SegmentNT (de Almeida et al., 2024) scaled models to billions of parameters. These models achieved promising results in understanding DNA genomes, with parameter counts reaching up to 2.5 billion and 1 billion, respectively. AgroNT, pre-trained on multi-species edible plant DNA sequences, however, did not transfer effectively to RNA sequence modeling in subsequent experiments. Addressing the challenge posed by the considerable length of genomic sequences, recent works have emphasized long-range sequence modeling and introduced auto-regressive FMs, such as HyenaDNA (Nguyen et al., 2023) and Evo (Nguyen et al., 2024).

In the context of RNA genomic modeling, several preliminary studies have emerged, including scBERT (Yang et al., 2022), RNABERT (Akiyama & Sakakibara, 2022), RNA-FM (Chen et al., 2022), RNA-MSM (Zhang et al., 2023), and RNAErnie (Wang et al., 2024). These models, however, are typically trained on limited-scale databases due to the scarcity and expense of obtaining RNA sequences. Some FMs focus on specific RNA types, such as coding sequences (CDS)(Hallee et al., 2023), 5' untranslated regions (5'UTR)(Chu et al., 2024), 3' untranslated regions (3'UTR)(Yang et al., 2023), or precursor mRNA sequences(Chen et al., 2023), which constrains their ability to capture the full diversity of RNA sequences. Uni-RNA (Wang et al., 2023) has been reported to achieve strong performance owing to its large-scale model and extensive database. However, it is

not open-sourced, precluding direct comparison in experiments. ChatNT (Richard et al., 2024) is a multimodal conversational agent designed to assist with tasks involving DNA, RNA, and protein sequences. It can handle diverse genomic and proteomic tasks, such as predicting sequence structures, simulating biological processes, or interacting with foundational models. ChatNT integrates advanced AI models to facilitate research in genomic data processing, enhancing accessibility and scalability in tasks across multiple biological modalities.

# B  BENCHMARK DETAILS

## B.1  RNA GENOMIC BENCHMARK

The detailed task descriptions for each nucleic acid and species, including the number of examples, classes, evaluation metric, and sequence length, are outlined in Table 5. Each task is carefully curated to reflect the complexity and variety inherent in genomic data, providing a robust framework for assessing the nuanced capabilities of state-of-the-art RNA FMs. RGB contains 6 SN-level tasks that are curated or collected from published articles. The purpose of RGB is to benchmark genomic FMs in challenging SN-level modeling tasks such as the detection and repair of SN mutations, mRNA sequence degradation rates, and RNA secondary structure prediction. Due to the lack of a plant RNA benchmark dataset, RGB includes the modeling of RNA sequences from a variety of species, e.g., plant and human. The sequence length in RGB ranges from 107 to 512, which is sufficient for most RNA understanding tasks. In summary, these multi-species and SN-level tasks in RGB serve as the first comprehensive benchmark utilized to assess the RNA sequence modeling capabilities of `GFMBench` and its baseline models. The brief introduction of the datasets in RGB is as follows:

- **Single-Nucleotide Mutation Detection (SNMD)**: We developed a plant RNA dataset synthesizing the single-nucleotide mutations. Focused on identifying potential single nucleotide changes, this task is essential for detecting mutations linked to genetic disorders. The SNMD dataset introduces up to 10 random mutations in the original sequences, regardless of variation ratios. Cross-entropy is utilized as the loss function for this binary token classification task.

- **Single-Nucleotide Mutation Repair (SNMR)**: This task challenges the model to suggest corrective actions at the single nucleotide level, aiding in gene therapy approaches. The SNMR dataset mirrors the SNMD dataset, with cross-entropy as the loss function, indicating a token 4-way (i.e., `A`, `U`, `C`, `G`) classification task.

- **mRNA Degrade Rate Prediction (mRNA)**: Estimating the decay rate of nucleotides in mRNA sequences, this task is vital for deciphering gene expression and regulation. The dataset originates from the Kaggle COVID-19 vaccine design competition[8], focusing solely on sequence-based degradation rate prediction and excluding RNA structures. It's a token regression task using MSE as the loss function, with the dataset re-split into training, validation, and testing sets for evaluation.

- **RNA Secondary Structure Prediction (bpRNA & Archive2 & RNAStralign)**: Aiming to predict RNA folding into secondary structures, this task is fundamental to RNA functionality and interactions. We evaluated `GFMBench` on four datasets, bpRNA (Danaee et al., 2018) (TR0, VL0, TS0 sets), ArchiveII (Mathews, 2019), RNAStralign (Tan et al., 2017) and Rfam (Kalvari et al., 2021). Following existing works, we have excluded sequences over 512 bases and complex structures, simplifying to three symbols: '`(`', '`.`', '`)`' Results may not directly compare with other studies due to these modifications. Cross-entropy serves as the loss function.

Please find the appendix for the input and output examples of each subtask in RGB. The detailed task descriptions for each nucleic acid and species, including the number of examples, classes, evaluation metric, and sequence length, are outlined in Table 5. Each task is carefully curated to reflect the complexity and variety inherent in genomic data, providing a robust framework for assessing the nuanced capabilities of state-of-the-art RNA FMs.

Table 6 show the virtual examples of different datasets in RGB. Please refer to our supplementary materials to find the datasets for more details.

---

[8]`https://www.kaggle.com/competitions/stanford-covid-vaccine`

Table 5: The brief statistics of subtasks in the RGB. These benchmark datasets are held out or not included in the pretraining database. The numbers of examples in training, validation and testing sets are separated by "/". * indicates the datasets are used for zero-shot performance evaluation only.

| Task | Task Type | # of examples | # of classes | Metric | Sequence length | Source |
|------|-----------|---------------|--------------|--------|-----------------|--------|
| SNMD | Token classification | $8,000/1,000/1,000$ | 2 | AUC | 200 | This work |
| SNMR | Token classification | $8,000/1,000/1,000$ | 4 | macro F1 | 200 | This work |
| mRNA | Token regression | $1,735/193/192$ | — | RMSE | 107 | Kaggle |
| bpRNA | Token classification | $10,814/1,300/1,305$ | 3 | macro F1 | $\leq 512$ | (Danaee et al., 2018) |
| AchiveII | Token classification | $2278/285/285$ | 3 | macro F1 | $\leq 500$ | (Mathews, 2019) |
| RNAStrAlign | Token classification | $17483/2186/2185$ | 3 | macro F1 | $\leq 500$ | (Tan et al., 2017) |

Table 6: The virtual input and output examples in the four benchmarks. The "..." represents the sequences that are omitted for better presentation and the red color indicates the wrong prediction in classification tasks. In the mRNA dataset, all single nucleotides have three values to predict. Note that "T" and "U" can be regarded as the same symbol in RNA sequences and depend on different datasets.

| Genome Type | Dataset | Column | Examples |
|-------------|---------|--------|----------|
| RNA | SNMD | Input Sequence | G A G T A ... T T G A G |
| | | True Label | 0 0 1 0 0 ... 0 0 1 0 0 |
| | | Prediction | 0 0 0 0 0 ... 0 0 1 0 0 |
| | SNMR | Input Sequence | T A C G A ... C T G A T |
| | | True Label | T A C A A ... G T A A T |
| | | Prediction | T A C A A ... C T G A T |
| | mRNA | Input Sequence | G G ... A C |
| | | True Label | [0.1,0.3,0.2] [0.8,0.4,0.1]... [0.9,0.4,0.3] [0.5,0.2,0.6] |
| | | Prediction | [0.1,0.3,0.2] [0.8,0.4,0.1]... [0.9,0.4,0.3] [0.5,0.2,0.6] |
| | bpRNA | Input Sequence | G G C G A ... C U U U U |
| | | True Label | ( ( ( · ·...· · ) ) ) |
| | | Prediction | ( ( ( ( ·...· ) ) ) ) |
| DNA | Classification | Input Sequence | A T C G A ... T A G |
| | | True Label | 1 |
| | | Prediction | 0 |
| | Regression | Input Sequence | G C C A T ... G C T |
| | | True Label | 2.56 |
| | | Prediction | 2.45 |
| | Chrom Acc (Multi-label) | Input Sequence | A T C G ... C T G |
| | | True Label | [1, 0, 1, 1, 0, 1, 1, 0, 1] |
| | | Prediction | [1, 1, 1, 1, 0, 1, 1, 0, 1] |

Table 7: The genomic tasks in the Plant Genomic Benchmark. This table briefly enumerates each task by name, the number of datasets available, the type of classification or regression analysis required, the range of sequence lengths, and the total number of samples in each dataset. Please find the dataset details of PGB in Agro-NT.

| Task | # of datasets | Task Type | Total # of examples | # of classes | Metric | Sequence length |
|------|---------------|-----------|---------------------|--------------|--------|-----------------|
| Polyadenylation | 6 | Sequence classification | $738,918$ | 2 | macro F1 | 400 |
| Splice site | 2 | Sequence classification | $4,920,835$ | 2 | macro F1 | 398 |
| LncRNA | 2 | Sequence classification | $58,062$ | 6 | macro F1 | $101 - 6000$ |
| Promoter strength | 2 | Sequence regression | $147,966$ | — | RMSE | 170 |
| Terminator strength | 2 | Sequence regression | $106,818$ | — | RMSE | 170 |
| Chromatin accessibility | 7 | Multi-label classification | $5,149,696$ | $9 - 19$ | macro F1 | $1,000$ |
| Gene expression | 6 | Multi-variable regression | $206,358$ | — | RMSE | $6,000$ |
| Enhancer region | 1 | Sequence classification | $18,893$ | 2 | macro F1 | $1,000$ |

## B.2 PLANT GENOMIC BENCHMARK

PGB (Mendoza-Revilla et al., 2023) provides a comprehensive suite of datasets designed to evaluate and improve the predictive capabilities of GFMs in plant biology. This benchmark, as shown in Table 7, encompasses a range of critical genomic tasks, including binary classification, single and multi-variable regression, and multi-label classification, addressing various aspects of plant genomics such as RNA processing, gene expression, and chromatin accessibility. By integrating diverse genomic tasks, the PGB aims to facilitate advanced research and development in plant genomics, offering a robust platform for the assessment and enhancement of model performance across different plant species. To obtain a detailed description of PGB, please refer to Agro-NT (Mendoza-Revilla et al., 2023).

### B.3 GENOMIC UNDERSTANDING EVALUATION

GUE (Zhou et al., 2023) serves as a DNA genomic benchmark, encompassing 36 datasets across nine crucial genome analysis tasks applicable to a variety of species. Similar to PGB and GB, it is used for evaluating the generalizability of `GFMBench` on DNA genome benchmarking. To thoroughly assess the capabilities of genome foundation models across sequences of varying lengths, tasks have been chosen with input lengths spanning from 70 to 10,000. The brief statistics for each dataset included in the GUE benchmark are displayed in Table 8, and the task descriptions are available in Zhang et al. (2023). Due to resource limitations, we do not include large-scale FMs in this benchmark, e.g., Agro-NT. Besides, we run the evaluation on a subset of GUE, where for each task we randomly select at most 10k samples from the original splits, e.g., training, testing and validation (if any) sets.

Table 8: Statistics of tasks in the GUE, these details can be found in Section B.2. from Zhang et al. (2023).

| Task | Metric | Datasets | Training | Validation | Testing |
|---|---|---|---|---|---|
| Core Promoter Detection | macro F1 | tata | 4,904 | 613 | 613 |
| | | notata | 42,452 | 5,307 | 5,307 |
| | | all | 47,356 | 5,920 | 5,920 |
| Promoter Detection | macro F1 | tata | 4,904 | 613 | 613 |
| | | notata | 42,452 | 5,307 | 5,307 |
| | | all | 47,356 | 5,920 | 5,920 |
| Transcription Factor Prediction (Human) | macro F1 | wgEncodeEH000552 | 32,378 | 1,000 | 1,000 |
| | | wgEncodeEH000606 | 30,672 | 1,000 | 1,000 |
| | | wgEncodeEH001546 | 19,000 | 1,000 | 1,000 |
| | | wgEncodeEH001776 | 27,497 | 1,000 | 1,000 |
| | | wgEncodeEH002829 | 19,000 | 1,000 | 1,000 |
| Splice Site Prediction | macro F1 | reconstructed | 36,496 | 4,562 | 4,562 |
| Transcription Factor Prediction (Mouse) | macro F1 | Ch12Nrf2\iggrab | 6,478 | 810 | 810 |
| | | Ch12Zrf384hpa004051\iggrab | 5,395 | 674 | 674 |
| | | MelJun\iggrab | 2,620 | 328 | 328 |
| | | MelMafkDm2p5dStd | 1,904 | 239 | 239 |
| | | MelNelf\iggrab | 15,064 | 1,883 | 1,883 |
| Epigenetic Marks Prediction | macro F1 | H3 | 11,971 | 1,497 | 1,497 |
| | | H3K14ac | 26,438 | 3,305 | 3,305 |
| | | H3K36me3 | 29,704 | 3,488 | 3,488 |
| | | H3K4me1 | 25,341 | 3,168 | 3,168 |
| | | H3K4me2 | 24,545 | 3,069 | 3,069 |
| | | H3K4me3 | 29,439 | 3,680 | 3,680 |
| | | H3K79me3 | 23,069 | 2,884 | 2,884 |
| | | H3K9ac | 22,224 | 2,779 | 2,779 |
| | | H4 | 11,679 | 1,461 | 1,461 |
| | | H4ac | 27,275 | 3,410 | 3,410 |
| Covid Variant Classification | macro F1 | Covid | 77,669 | 7,000 | 7,000 |
| Enhancer Promoter Interaction | macro F1 | GM12878 | 10,000 | 2,000 | 2,000 |
| | | HeLa-S3 | 10,000 | 2,000 | 2,000 |
| | | HUVEC | 10,000 | 2,000 | 2,000 |
| | | IMR90 | 10,000 | 2,000 | 2,000 |
| | | K562 | 10,000 | 2,000 | 2,000 |
| | | NHEK | 10,000 | 2,000 | 2,000 |
| Species Classification | macro F1 | fungi | 8,000 | 1,000 | 1,000 |
| | | virus | 4,000 | 500 | 500 |

### B.4 GENOMIC BENCHMARKS

GB is also a DNA-oriented FM benchmark suite, which can be used for generalizability evaluation of OmniGenome. It contains a well-curated collection of datasets designed for the classification of genomic sequences, focusing on regulatory elements across multiple model organisms. This collection facilitates robust comparative analysis and development of genomic FMs. The task names in the original repository are complex, we abbreviate the names as follows:

- DEM corresponds to "Demo Coding vs Intergenomic Seqs"
- DOW is for "Demo Human or Worm"

- DRE represents "Drosophila Enhancers Stark"

- HCE is short for "Human Enhancers Cohn"

- HEE denotes "Human Enhancers Ensembl"

- HRE abbreviates "Human Ensembl Regulatory"

- HNP shortens "Human Nontata Promoters"

- HOR is an abbreviation for "Human Ocr Ensembl"

- DME simplifies "Dummy Mouse Enhancers Ensembl"

The brief statistics for each dataset included in the GUE benchmark are displayed in Table 8. Similar to GUE, we run the evaluation on a subset of GB, where for each task we randomly select at most 10k samples from the original splits, e.g., training, testing and validation (if any) sets.

Table 9: The brief statistics of datasets reported in the genomic benchmark (Grešová et al., 2023).

| Task | # of Sequences | # of Classes | Class Ratio | Median Length | Standard Deviation |
|------|----------------|--------------|-------------|---------------|--------------------|
| DME | $1,210$ | 2 | 1.0 | $2,381$ | 984.4 |
| DEM | $100,000$ | 2 | 1.0 | 200 | 0.0 |
| DOW | $100,000$ | 2 | 1.0 | 200 | 0.0 |
| DRE | $6,914$ | 2 | 1.0 | $2,142$ | 285.5 |
| HCE | $27,791$ | 2 | 1.0 | 500 | 0.0 |
| HEE | $154,842$ | 2 | 1.0 | 269 | 122.6 |
| HRE | $289,061$ | 3 | 1.2 | 401 | 184.3 |
| HNP | $36,131$ | 2 | 1.2 | 251 | 0.0 |
| HOR | $174,456$ | 2 | 1.0 | 315 | 108.1 |

## C    DATA FILTERING IN BENCHMARKING

The pertaining involves RNA sequences and structures prediction, we take the data and annotation leakage problem seriously.

- To avoid structure annotation leakage of downstream benchmarks, the secondary structure predictors for all FMs were randomly initialized for fair comparisons, which means the pre-trained structure predictor of `GFMBench` was not used in benchmarks, except for zero-shot SSP experiments. Please find the source codes for details.

- To reduce sequence leakage caused by evolutionary conservative sequences across multiple species, we use the ch-hit-est tool to calculate the sequence similarity between sequences from the OneKP database and downstream tasks. We adopt the similarity threshold of $80\%$ for ch-hit-est (Li & Godzik, 2006) to eliminate sequences whose homogeneous sequences appeared in the OneKP database. Subsequently, we exploit the blastn (Altschul et al., 1990) tool to query potentially leaked sequences in downstream benchmark datasets and further alleviate the data leakage problem. The e-value has been set to 1 for rigorous sequence filtering.

### C.1    EXPERIMENT SETTINGS

In this experiment, we carefully selected a set of key hyperparameters to optimize model performance. Below are the main hyperparameter settings along with detailed explanations:

- **Dropout**: To prevent the model from overfitting during training, we set the Dropout value to 0, meaning that no random neuron dropout is applied during training. This choice was made based on our consideration of model stability and generalization ability.

- **Learning Rate**: We set the learning rate to 2e-5, which is a relatively small value to ensure stable convergence, especially in complex training tasks. A smaller learning rate helps to avoid drastic fluctuations during the training process, leading to more precise optimization.

Table 10: The brief statistics of RNA and DNA FM baselines. Please note that the pertaining data scales cannot be directly compared because the measurements are different in various publications. The detailed introduction of these FMs can be found in original publications.

| Model | Tokenization | # of Params. | Pre-training Data Scale | Pre-training Data Source | Species | Sequence Type |
|---|---|---|---|---|---|---|
| DNABERT-2 | BPE | 117M | 32.49B Tokens | The 1000 Genomes Project | Human + 135 Species | DNA |
| NT-V2-100M | k-mers | 96M | 300B Tokens | The 1000 Genomes Project, etc. | Human + 850 Species | DNA |
| HyenaDNA-Large | SNT | 47M | 3.2B Tokens | Genome Reference Consortium | Human | DNA |
| Caduceus | SNT | 1.9M | 35B Tokens | Genome Reference Consortium | Human | DNA |
| Agro-NT-1B | k-mers | 985M | 472.5B Tokens | Ensembl Plants Database | 48 Edible Plants | DNA |
| SpliceBERT | SNT | 19M | 2M Sequences | UCSC Genome Browser | Multi-Vertebrates | precursor-mRNA |
| RNA-BERT | SNT | 0.5M | 4, 069 RNA Families | The RNA Families Database | Multi-Species | ncRNA |
| RNA-MSM | SNT | 96M | 4, 069 RNA Families | The RNA Families Database | Multi-Species | ncRNA |
| RNA-FM | SNT | 96M | 23M Sequences | RNAcentral Database | Multi-Species | ncRNA |
| 3UTRBERT | k-mers | 86M | 20, 362 Sequences | The GENCODE Project | Human | mRNA 3'UTR |
| OmniGenome | SNT | 186M | 54.2B Tokens | The OneKP Initiative | 1124 Plant Species | mRNA, CDS, UTR |

- **Weight Decay**: We applied a weight decay of 0.01 to control model complexity and prevent overfitting. Weight decay is a regularization technique that effectively constrains the growth of model parameters, maintaining the model's generalization capability.

- **Adam Optimizer**: We used the Adam optimizer with its parameters set to $\beta_1 = 0.9$ and $\beta_2 = 0.999$. The Adam optimizer combines the benefits of momentum and adaptive learning rates, accelerating convergence and adapting to different gradient changes, thereby improving the efficiency and effectiveness of model training.

- **Learning Rate Scheduler**: We opted for a linear decay learning rate scheduler, allowing the learning rate to gradually decrease during training. This strategy helps the model make smaller adjustments as it approaches the optimal solution, ensuring a better convergence outcome.

- **Batch Size**: The batch size was set to 8. This relatively small batch size helps to efficiently train the model within limited memory resources, particularly when handling large-scale data, enabling a balance between model performance and computational resource usage.

- **# of Epochs**: We set the number of training epochs to 20. This setting ensures that the model can fully learn the features within the data while avoiding the negative effects of overtraining.

- **Early Stopping**: We implemented an early stopping mechanism, terminating the training early if the validation performance does not improve for 5 consecutive epochs. This mechanism effectively prevents model overfitting and saves training time.

It is important to note that for different tasks, some hyperparameter settings may be adjusted. To obtain accurate experimental results, please refer to the detailed parameter configurations in the compiled dataset specific to each task.

## C.2 DEVELOPMENT ENVIRONMENT

The benchmark experiments based on `GFMBench` were conducted on a dedicated Linux computation node, equipped with 2 NVIDIA RTX 4090 GPUs. For distributed model training, we employed version 4.44.0 of the Transformers library alongside version 0.28.3 of the Accelerate library. Our implementation framework of choice for `GFMBench` was PyTorch, specifically version 2.1.0. The ViennaRNA version is 2.6.4 in our experiments. While some existing code was adapted for the modules within `GFMBench`, the majority of the codebase, such as genomic sequences preprocessing, model pre-training, objective functions, and experiments, was meticulously crafted from scratch.

## C.3 EVALUATION BASELINES

To comprehensively evaluate the performance of the existing GFMs across the integrated benchmarks, i.e., RGB, PGB, GUE and GB, we have obtained the results of existing GFMs based on `GFMBench`.

Please note that it is assumed that the structure annotation from ViennaRNA is always available for structure-contextualized modeling to enhance OmniGenome. In SSP tasks, we can also use the ViennaRNA's structure annotations as contexts to improve downstream SSP performance. Please refer to Appendix C.3 for brief introductions of these FMs.

We can compare `GFMBench` with the following RNA and DNA FMs shown in Table 10 as baselines to help evaluate the performance of `GFMBench`. We are aware that some FMs are also developed for RNA, such as Uni-RNA (Wang et al., 2023), 5UTR-LM (Chu et al., 2024), etc. However, we cannot compare `GFMBench` with them because their source codes are very hard to work with in our efforts or are not publicly available. To help understand the baseline FMs, we briefly summaries the FM in the following sections. Please find the method and experiment details of these FMs in the original publications.

- ViennaRNA (Lorenz et al., 2011). ViennaRNA is a comprehensive genomic analysis tool that includes a diverse set of interfaces, such as RNAFold[9] and RNAInverse[10] design. ViennaRNA serves as the baseline for RNA structure prediction and RNA design in our experiments.

- DNABERT2 (Zhou et al., 2023). DNABERT2 is one of the latest DNA FMs which improves the performance of DNABERT. The main modification of DNABERT2 is the tokenization method, which was changed to BPE from k-mers.

- HyenaDNA (Nguyen et al., 2023). HyenaDNA is an autoregressive FM optimized for long-range genome data processing. HyenaDNA is based on the Hyena convolution architecture and capable of handling sequences up to 1M bases in length.

- Caduceus (Schiff et al., 2024a). Caduceus[11] is an advanced DNA language model built on the MambaDNA architecture, designed to address challenges in genomic sequence modeling, such as long-range token interactions and reverse complementarity (RC).

- Nucleotide Transformer (NT) V2 (Dalla-Torre et al., 2023). The NT FMs were trained on DNA data, including the human reference genome and multi-species DNA sequences. They aim to capture the complex patterns within nucleotide sequences for various genome modeling applications.

- Agricultural Nucleotide Transformer (Agro-NT) (Mendoza-Revilla et al., 2023). Agro-NT is a large-scale DNA FM (1B parameters) akin to the Nucleotide Transformers but with a focus on plant DNA.

- SpliceBERT (Chen et al., 2023). It was trained on 2M precursor messenger RNA (pre-mRNA) and specialised in RNA splicing of pre-mRNA sequences.

- 3UTRBERT (Yang et al., 2023). This model was trained on 20k 3'UTRs for 3'UTR-mediated gene regulation tasks. It uses k-mers tokenization instead of SNT. RNA-BERT (Akiyama & Sakakibara, 2022). RNA-BERT is a BERT-style model pre-trained on a large corpus of non-coding RNA sequences. It uses masked language modeling (MLM) as its primary training objective. The model is designed to predict RNA structural alignments and can be fine-tuned for various RNA sequence classification and regression tasks

- RNA-MSM (Zhang et al., 2024) RNA-MSM is an unsupervised RNA language model based on multiple sequence alignment (MSA). It is the first model of its kind to produce embeddings and attention maps that directly correlate with RNA secondary structure and solvent accessibility. RNA-MSM is particularly effective for tasks involving evolutionary relationships in RNA sequences.

- RNA-FM (Chen et al., 2022) RNA-FM is a BERT-based RNA foundation model trained on a vast dataset of non-coding RNA sequences. The model excels in predicting RNA structure and function by leveraging masked language modeling (MLM) during pre-training. RNA-FM's training data is sourced from the RNAcentral database, providing it with extensive knowledge across diverse RNA species.

- `GFMBench`. `GFMBench` is the RNA genome FM that advocates the importance of sequence-structure alignment. Moreover, it is the first FM which addressed the *in-silico* RNA design task.

- **OmniGenome**: A FM dedicated to RNA genome modeling. This model leverages the computation-based structure to enhance the genome modeling ability and archives impressive performance on both RNA and DNA genomes.

---

[9]https://www.tbi.univie.ac.at/RNA/RNAfold.1.html
[10]https://www.tbi.univie.ac.at/RNA/RNAinverse.1.html
[11]https://huggingface.co/kuleshov-group/caduceus-ps_seqlen-131k_d_model-256_n_layer-16

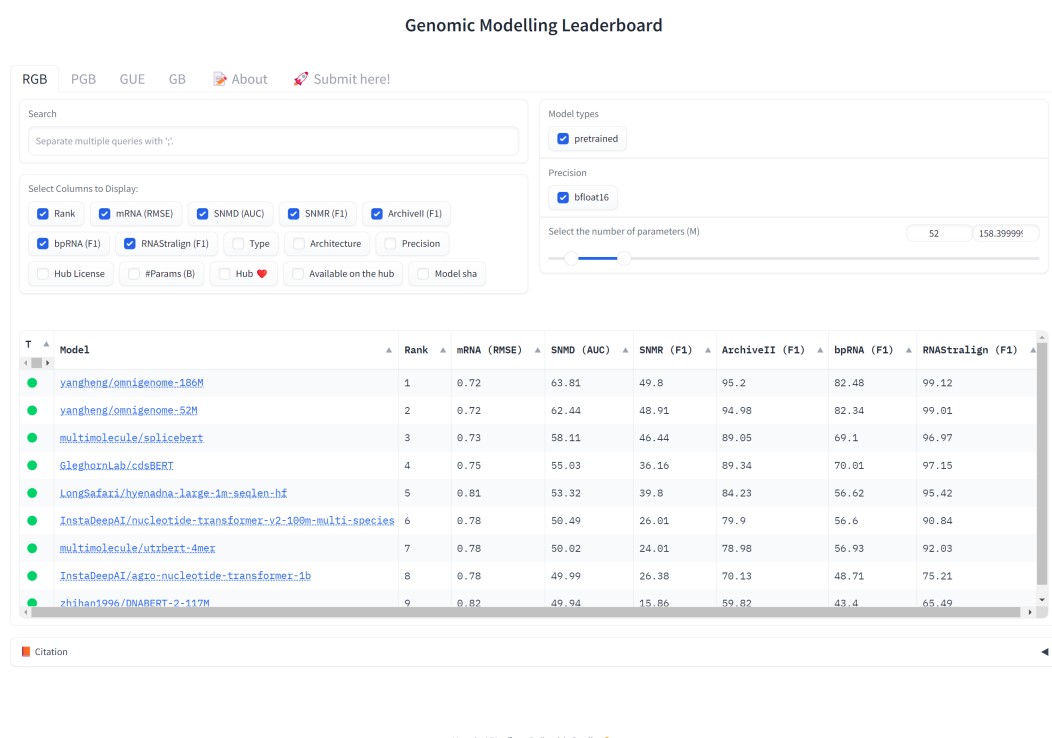

Figure 2: The current webpage interface of the public leaderboard.

## D    PUBLIC LEADERBOARD

The public leaderboard has been launched with the manuscript, and the current layout of the leaderboard is illustrated in Figure 2. We have included the results of open-source GFMs among four benchmark suites, and new results can be expected from the community. We are still working to include the performance of recent GFMs, and refine the leaderboard interface with better integrity.

## E    LIMITATIONS

The GFM benchmarking may not reflect the accurate performance in biology reality, we attribute the limitations of benchmarking to two major aspects:

- **Lack of *in-vivo* Data**: One of the critical limitations of GFMs lies in the absence of *in-vivo* verified genome data. While GFMs perform well in *in-silico* environments, where computational models and simulations are used to predict biological processes, these models are rarely validated against *in-vivo* data, which refers to experimental data obtained from living organisms. This presents a significant challenge for accurately translating model predictions to real-world biological applications. To be more specific, the complexity of biological systems, including interactions within cells, tissues, and organisms, often introduces variables that are not fully captured in computational simulations. For example, gene regulation, environmental factors, and cellular responses to genetic modifications may behave differently in living organisms than predicted by models trained on *in-silico* data. As a result, GFMs might not fully capture the biological complexity, leading to discrepancies between predicted and actual outcomes.

- **Model Scale Constraints**: The second major limitation is the model scales in benchmarking. As GFMs become larger and more sophisticated, their performance improves, but this scaling comes at a significant cost. Training as well as benchmarking large-scale GFMs requires immense computational resources, including high-performance GPUs or TPUs, massive memory allocation, and extensive storage for datasets. The cost of acquiring and maintaining this infrastructure can be prohibitive for many research institutions or companies, limiting access to cutting-edge GFMs.

## F ETHIC STATEMENT

The development of GFMs presents various ethical challenges that must be carefully considered. As we push the boundaries of what is possible with large-scale GFMs, such as Evo, it is crucial to establish a responsible framework for their development and application. GFMs enable advanced capabilities like generating and predicting DNA sequences at a whole-genome scale, which opens the door to significant breakthroughs in fields such as genetic engineering and therapeutic development. However, these same capabilities pose risks related to bio-security, inequality, and environmental disruption.

Safety and Ethical Implications: GFMs like OmniGenome could be misused by malicious actors for harmful purposes, such as creating synthetic organisms that could threaten bio-safety. It is essential to establish strict guidelines on access and use, including the development of safety guardrails, access controls, and audits to monitor queries and research outcomes.

Health and Social Inequity: While the open-source nature of GFMs promotes transparency and accessibility, there are concerns that the benefits of these tools may disproportionately favor well-resourced organizations, such as pharmaceutical companies, which could lead to further inequalities in global health. Intellectual property considerations also arise, as companies using open-source tools might monopolize treatments or set prohibitive costs, exacerbating health disparities.

Environmental Impact: The enhanced capabilities for genetic manipulation that GFMs enable could disrupt natural ecosystems, leading to potential loss of biodiversity or the emergence of harmful species. Additionally, the computational demands of training large models have environmental costs, such as increased carbon footprints, that must be weighed against the benefits of the scientific advancements.

In response to these concerns, we are committed to promoting ethical guidelines, transparency, and the responsible use of GFMs. We will collaborate with the community to continually refine these guidelines as the field evolves.

## G SOCIAL IMPACT

The societal impact of GFMs is substantial, with applications ranging from personalized medicine to environmental management. These models have the potential to revolutionize fields such as healthcare and agriculture by providing deeper insights into genetic data, enabling the discovery of new biomarkers, and assisting in the development of more effective therapies. In healthcare, GFMs can drive advancements in precision medicine, allowing for personalized treatments based on individual genetic profiles, which could drastically improve patient outcomes for conditions such as cancer or rare genetic disorders. In agriculture, GFMs can contribute to sustainable practices by improving crop yields and resistance to disease. However, careful consideration must be given to the ecological balance, as genetic modifications could have unforeseen consequences on ecosystems. As GFMs continue to evolve, their responsible development and deployment will be crucial to ensuring that their societal impact is positive and equitable.

However, there are also risks associated with the unequal access to these powerful tools. Entities with more resources and technical expertise may benefit disproportionately from GFMs, accelerating their research and economic returns while leaving lower-resourced institutions and countries at a disadvantage. To mitigate this, it is critical to ensure that access to GFMs is democratized through open-source initiatives, global collaboration, and capacity-building efforts in low-resource settings.

