# OpenReview forum: "Automating Large-scale In-silico Benchmarking for Genomic Foundation Models"
_ICLR.cc/2025/Conference — Submitted to ICLR 2025_

### Official Review · Reviewer_MGAx · 2024-10-31

**Soundness:** 2
**Presentation:** 2
**Contribution:** 1
**Rating:** 1
**Confidence:** 4

**Summary:**

This paper introduces GFMBench, a framework for standardizing and automating benchmarking of Genomic Foundation Models (GFMs), which integrates four large-scale benchmarks containing 42 million genomic sequences across 75 datasets. The framework addresses several challenges in genomic modeling by providing unified protocols, consistent metrics, and automated evaluation pipelines that work across both DNA and RNA tasks from multiple species. GFMBench is released as open-source software with user-friendly interfaces, built-in tools for genomic tasks, and includes a public leaderboard and online hub for sharing benchmark results and resources with the research community.

**Strengths:**

The paper integrates 4 large genomics benchmarks from other groups into a unified framework. They also create a way to run a pipeline with these benchmarks with GFMs.

**Weaknesses:**

The paper exhibits several significant limitations. First, the technical innovation appears limited in scope. While the work integrates multiple existing benchmarks into a unified framework, it does not create new benchmarking methodologies or metrics. The automation and standardization approaches described largely follow established machine learning practices, without introducing fundamental methodological advances in benchmarking or evaluation techniques.

The work's focus is predominantly engineering-oriented rather than research-driven. The main contributions center on developing practical tools and interfaces for the genomics community. While valuable from a practical perspective, the emphasis on standardization over novel scientific insights raises questions about its suitability for a top-tier machine learning conference. Furthermore, similar benchmark aggregation platforms already exist in other machine learning domains, making this approach less innovative.

**Questions:**

What specific technical innovations in benchmarking does GFMBench introduce?

What novel metrics or evaluation approaches are introduced?

What specific reproducibility issues does it solve that other frameworks don't?

Why were these 4 benchmarks and then GFMs selected among others?

In Table 4, why is GFMBench listed as a model?

---

> ### Author Response · Authors · 2024-11-20
> **Clarifications and Rebuttal**
>
> **Weakness**
>
> **First, the technical innovation appears limited in scope.**
> **Reply:** The primary objective of this work is to develop a platform solution that addresses the critical need for a universal genomic benchmarking and application toolbox. This effort is aimed at standardizing and automating the benchmarking process, particularly for RNA and DNA tasks, to address reproducibility challenges often encountered with GFMs.
> Moreover, our package also features creating an accessible, ready-to-use framework for bioinformatics applications, as demonstrated in the tutorials (e.g., RNA design, RNA augmentation, and embedding extraction). While we do not introduce new benchmarking methodologies or metrics, the platform supports custom metrics, models, datasets, and pipelines to ensure flexibility and adaptability.
>
> **While the work integrates multiple existing benchmarks into a unified framework, it does not create new benchmarking methodologies or metrics.**
> **Reply:** The new metrics are often problem-specific and may require targeted development based on specific applications, which are out of the scope of this platform. Our emphasis in this work has been on providing a universal and robust benchmarking foundation that can be extended and refined over time. The four benchmark suites included cover a broad range of genomic data types (RNA and DNA) and species diversity (plants, humans, and bacteria), making this an important step towards comprehensive genomic benchmarking.
>
> **The automation and standardization approaches described largely follow established machine learning practices, without introducing fundamental methodological advances in benchmarking or evaluation technique.**
> **Reply:** Solutions always matter, especially for the domains that are emerging and facing the reality of lacking such resources. While the platform is designed to evolve incrementally with iterative updates, we believe the current contribution lays a strong foundation for standardizing genomic benchmarks and enabling practical applications, and thus does not represent a substantial limitation of this work.
>
> **The work's focus is predominantly engineering-oriented rather than research-driven. The main contributions center on developing practical tools and interfaces for the genomics community. While valuable from a practical perspective, the emphasis on standardization over novel scientific insights raises questions about its suitability for a top-tier machine learning conference.**
> **Reply:** The rationale and prospective benefits of genomics research are discussed in the introduction section, e.g., the knowledge transfer evaluation via adaptive benchamrking.
> - While it is true that our work is primarily engineering-oriented, we believe standardization and automation are crucial for advancing the AI4Genomics community.  These aspects address fundamental challenges in the field, as demonstrated by their transformative impact in other areas, such as language modeling.
> - The genomics research currently lacks a universal platform to address reproducibility issues in model evaluation and to lower the entry barrier for leveraging GFMs in genomic applications.  Without such a platform, researchers, particularly novices, face significant challenges in applying these models effectively and reproducibly. Our work focuses on filling this gap by providing a standardized, automated, and extensible framework tailored to the unique demands of genomics.
> - While benchmark aggregation platforms exist in other domains, our work specifically adapts these principles to the genomic context, which presents distinct challenges, including diverse data formats, sequence types, and evaluation criteria.  We believe that this platform-oriented contribution is highly valuable for the community, as it accelerates development, facilitates reproducibility, and broadens access to GFMs.
> - In summary, while the emphasis of this work is on practical solutions, it addresses pressing and foundational needs in the genomics field, making it a meaningful contribution to the intersection of machine learning and genomics.

---

> > ### Author Response · Authors · 2024-11-20
> >
> > **Questions and Responses:**
> >
> > **1. What specific technical innovations in benchmarking does GFMBench introduce?**
> >
> > **Reply:** GFMBench unifies the GFM benchmark pipeline, enabling pre-configuration of all benchmarks. This approach is distinct because the configuration files include original metadata for datasets and detailed benchmarking settings, minimizing biases caused by unstandardized manual operations. Additionally, we have implemented support for custom models, tokenizers, datasets, and metrics (including pseudo-metrics). This flexibility allows users to compile and integrate new benchmarks independently, similar to the custom model loading functionality in libraries like Transformers. Finally, we implement adaptive benchmarking for the transfer evaluation between RNA and DNA models and benchmarks. This is an important feature as recent works indicate the DNA and RNA models may have generalizability to different modal data according to the in-silicon benchmarks.
> >
> > **2. What novel metrics or evaluation approaches are introduced?**
> >
> > **Reply:** GFMBench does not introduce novel metrics as we have no specific demands in this work and it falls outside the scope of our motivation, i.e., platform development. However, the framework supports the integration of custom metrics, enabling researchers to implement and evaluate new metrics as needed.
> >
> > **3. What specific reproducibility issues does it solve that other frameworks don't?**
> >
> > **Reply:** GFMBench addresses reproducibility by standardizing not only the datasets but also the benchmarking settings. Key hyperparameters such as random seeds, sequence lengths, and batch sizes are pre-defined in configuration files, ensuring consistent results and eliminating biases from implementation variations. Additionally, the evaluation components (e.g., metrics) are fixed and automatically loaded, avoiding discrepancies caused by metric implementation differences across experiments.
> >
> > **4. Why were these 4 benchmarks and then GFMs selected among others?**
> >
> > **Reply:** The selection was guided by the principles of diversity and coverage within the constraints of the time budget.
> > - **RGB:** Designed for fine-grained RNA model benchmarking.
> > - **PGB:** Tailored for plant DNA model benchmarking.
> > - **GUE:** Covers multiple species for DNA model benchmarking, including humans, bacteria, fungi, and more.
> > - **GB:** A widely-used benchmark in existing works, ensuring compatibility with most existing models for comparison. These benchmarks collectively provide broad genomic representation across different data types and species.
> >
> > **5. In Table 4, why is GFMBench listed as a model?**
> >
> > **Reply:** This was a typographical error and will be corrected in the revision. The entry should refer to OmniGenome, not GFMBench.

---

### Official Review · Reviewer_usCL · 2024-11-02

**Soundness:** 3
**Presentation:** 4
**Contribution:** 3
**Rating:** 5
**Confidence:** 3

**Summary:**

This paper introduces GFMBench, a framework for genomic foundation model (GFM)
evaluation. The proposed benchmark covers a diverse set of genomic tasks and datasets, and
provides methods for standardised metrics and evaluation, thus
enabling fair comparison of different GFMs. The papers include an evaluation of
multiple GFMs on the benchmark of standardised metrics, which allows the
comparison of the models. GFMBench is released as open-source software and
it includes an online leaderboard.

**Strengths:**

Generally, the paper is well-written in good English and easy to follow.
The motivation for the paper is clear and the problem is well defined.
Overall, the authors make a significant contribution to the field of genomics and
foundational genomics models. Providing a unifying benchmark for evaluation
is a significant contribution and will very much contribute positively to the
development of the progress of GFMs. The authors will release a webpage with a
leaderboard for model comparison, which is really positive. The authors
will also release the code for GFMBench.

**Weaknesses:**

There are some issues regarding the clarity of the contribution of
the paper and the reported results. The paper also presents some claims
that are not backed up or briefly supported. The issues are listed below:

Major issues:

- Data scarcity is mentioned to be an issue for genomics datasets in the paper.
While it is true that diversity is a problem in the access to genomic data it is
important to note that there are indeed many large-scale datasets available.
The main concern regarding this issue is that the authors claim that
GFMBench tackles this problem, however, as far as I'm concerned, the benchmark
does not contribute any novel data, but it is rather a compilation of
existing datasets. The claim that GFMBench mitigates "issues of data scarcity"
does not seem truly accurate.

- On page 5 it is claimed that "GFMBench" tackles the problem of
"benchmark standardization". While I understand the contribution of common
metrics and implementation, I don't see how this is achieved in the
"hyperparameter settings". How is standardization achieved in this regard?

- In Line 486, it is said that "OmniGenome consistently performs well across various genomic benchmarks",
however, Table 4 results do not seem to match this claim (correct me if I'm wrong).

- Through the paper, the authors praise the performance of "OmniGenome" GFM,
as it shows good performance, especially on RGB and PGB tasks.
Is OmniGenome also a contribution to the paper? If so, it would be great
to have more details on it. If not, it would be great to provide more insights
into the other models and why they perform better or worse in some tasks.
Similarly, since the contribution of the paper is a benchmark, it would be
more significant to provide some insight into the difficulty of the tasks
and why performance varies across them, instead of commenting only on OmniGenome.
I acknowledge that the paper highlights repeated times that the major performance
of OmniGenome is due to the fact that it is trained on "secondary structure" tasks,
However, this fact is repeated multiple times in the paper while no comment
is provided on the other models or tasks at hand and why "secondary structure"
improves performance on the tasks.

**Questions:**

A list of questions, minor issues and suggestions is included here:


- The abstract mentions "the absence of open-source software for diverse genomics".
This doesn't sound true to me. Genomics is especially rich in open-source
tools (e.g. see the collection of tools on the Galaxy platform
"The Galaxy platform for accessible, reproducible and collaborative biomedical analyses").

- On page 2 "lack of comprehensive and diverse datasets necessary for robust training and testing of GFMs.".
It would be great to support these claims with related work or references.

The first paragraph of page 4 is repeated twice.

- In Table 7 caption: "in Agro-NT".
Please include a citation to a tool, work, dataset or reference when mentioning
it for the first time. This happens also later in the text, e.g. "ViennaRNA",
"Archive2", "Stralign", etc.

- In Table 5 caption: "These benchmark datasets are held out or not included in the pretraining database".
It is unclear to me what this means. Would be thankful if the authors
could clarify what this means.

- What experiment is Appendix C referring to? Please clarify what these
hyperparameters are for.

- On page 6, "We provide a tutorial for AutoBench in ...".
To make the paper more consistent, it would be better to include the
tutorial URL in a footnote too.

- I understand that AutoBench is a standardisation suite for benchmarking, but
I believe that the paper could benefit from a more detailed explanation of it
and its contribution. For instance, how is it an "automated" benchmarking solution?

- Genome data augmentation is mentioned as a contribution, however,
no technical details are provided on how this is achieved. This is indeed
a subject of interest, and it would be great to have more
details.

- While I think that this is not a trivial issue, the paper does not mention
how are long sequences treated with the models. Is this because this is not an issue
for the model's evaluation? Otherwise, how is this managed? Is this handled or
should be handled in the GFMBench framework?

- Why does "GFMBench" appear as a model in Table 4?

- There are some spelling errors, e.g. "pertaining" in Line 1080.
Please double-check for spelling errors.

- "blastn" in Line 1066 is usually written in uppercase.

---

> ### Author Response · Authors · 2024-11-20
> **Clarifications and Rebuttal**
>
> **Weakness**
>
> **W1 Reply:** Thank you for raising this concern. The GFMBench is necessary because:
> - While it is true that large-scale genomic datasets are available, many of these datasets are not readily usable for benchmarking due to the significant effort required to parse and prepare data from heterogeneous formats. GFMBench does not aim to introduce novel data but rather focuses on addressing the accessibility and usability challenges by compiling, standardizing, and integrating existing datasets into a universal and adaptive benchmarking framework.
> - Our primary contribution lies in substantially lowering the barriers for researchers by providing a standardized, ready-to-use benchmarking suite. This enables efficient and consistent evaluation across diverse genomic tasks, mitigating practical issues associated with data preparation and compatibility. While GFMBench may not directly resolve the issue of data scarcity, it leverages existing comprehensive benchmarking to monitor related defects in the model pertaining.
>
> **W2 Reply:** Thank you for your question regarding standardization in hyperparameter settings.
> - GFMBench achieves this through the use of comprehensive configuration files for each benchmark, which are designed to ensure reproducibility and consistency.
> - These configuration files provide custom code support, enabling all metrics to be custom-implemented and fixed across evaluations. For hyperparameter settings, we standardized key parameters such as batch size, sequence length, and random seeds, ensuring that these remain consistent across all benchmarks. This uniformity eliminates variability introduced by differing hyperparameter choices, allowing for fair and reproducible comparisons.
> - In summary, the configuration files contain all the necessary information for reproducibility, and these settings are automatically loaded and enforced by AutoBench, ensuring standardized and transparent benchmarking.
>
> **W3 Reply:** The claim that "OmniGenome consistently performs well across various genomic benchmarks" reflects its ability to deliver competitive performance, even without being explicitly trained on human or other species data. While it may not achieve state-of-the-art results in every scenario, its performance remains comparable to models specifically trained for these tasks, demonstrating its generalizability. We acknowledge that the phrasing in the manuscript could be clearer, and we will revise it to better reflect this nuance, emphasizing that OmniGenome performs "competitively" rather than implying it is the state-of-the-art model across all DNA benchmarks.
>
> **W4 Reply:** Thank you for your observation and suggestion. To clarify, the models themselves are not the contribution of this work. However, our efforts in adapting and ensuring compatibility with the benchmarking framework can be seen as part of our contributions. We acknowledge that the current manuscript places considerable focus on OmniGenome. Based on your feedback, we will expand the discussion to include insights into all baseline models used in the benchmarking. This will include an analysis of why certain models perform better or worse on specific tasks and a deeper exploration of the task difficulties and their influence on performance variations.
> Additionally, we will provide more context on why "secondary structure" training improves performance on some tasks and discuss its implications alongside other factors influencing model performance. Thank you for highlighting this point, we will revise the manuscript accordingly to present a more balanced and comprehensive analysis.

---

> > ### Author Response · Authors · 2024-11-20
> > **Answers to Questions**
> >
> > **Abstract:**
> > **Reply:** Thank you for pointing this out. The research scope of our work is genomic foundation models (GFMs), specifically language models. We will revise our statement to reflect this suggestion.
> >
> > **Page 2, Data Scarcity Claim:**
> > **Reply:** We appreciate this suggestion. While this is a general observation about the field, it lacks specific references. We will revise this statement by summarizing and discussing relevant literature to provide more context and support.
> >
> > **Repetition on Page 4:**
> > **Reply:** Thank you for noting this oversight. We will rectify the duplication in the revision.
> >
> > **Table 7, Citations for Tools:**
> > **Reply:** We generally follow the principle of citing tools, datasets, or references upon first mention. However, we will carefully proofread the manuscript to ensure all instances, including those in Table 7 (e.g., "Agro-NT," "ViennaRNA," "Archive2," "Stralign"), are appropriately cited.
> >
> > **Table 5, Clarification on "Held Out":**
> > **Reply:** This statement addresses the issue of data leakage. Specifically, the datasets listed in Table 5 are not included in the pretraining database, ensuring the model does not encounter evaluation data during pretraining. We will clarify this in the text for better understanding.
> >
> > **Appendix C, Hyperparameters:**
> > **Reply:** These hyperparameters are commonly used for training neural models. To enhance clarity, we will elaborate on their definitions and how they relate to the training and evaluation processes in the manuscript.
> >
> > **Page 6, Tutorial URL:**
> > **Reply:** To ensure consistency, we will include the tutorial URL for AutoBench as a footnote in the revised manuscript.
> >
> > **AutoBench Explanation:**
> > **Reply:** AutoBench automates benchmarking by loading data, configurations, and benchmark settings from unified protocols. This eliminates the need for manual adjustments when handling different tasks, datasets, metrics, and models. We will expand the description in the manuscript to better explain its contribution and automated nature.
> >
> > **Genome Data Augmentation:**
> > **Reply:**  We implemented genome data augmentation using masked language modeling techniques. Technical details and examples are included in the supplementary materials. We will highlight this implementation more explicitly in the manuscript.
> >
> > **Long Sequence Processing:**
> > **Reply:** In GFMBench, sequences are used as-is without additional preprocessing for length. This ensures fair comparisons across models and avoids interference with their internal mechanisms. We believe this approach aligns with our goal of standardizing evaluations, but we will make this rationale explicit in the text.
> >
> > **Table 4, GFMBench as a Model:**
> > **Reply:** This is an oversight; the entry should refer to OmniGenome. We will correct this in the revision.
> >
> > **Spelling Errors and Typos:**
> > **Reply:** Thank you for pointing out the spelling error ("pertaining" on Line 1080) and the incorrect capitalization of "blastn" on Line 1066. We will carefully proofread the manuscript to address these and any other typographical issues.
> >
> > We are sorry for the text wall and appreciate your careful review of our work.

---

### Official Review · Reviewer_9soC · 2024-11-04

**Soundness:** 2
**Presentation:** 2
**Contribution:** 1
**Rating:** 3
**Confidence:** 3

**Summary:**

In this paper, the authors present an integrated platform, GFMBench, for benchmarking genomic foundation models. The authors synthesized and standardized four existing benchmarks and created a unified software framework to run GFMs and calculate evaluation metrics in a streamlined manner. In general, I consider this work lacks novelty and does not contribute sufficiently to the field, which is my primary reason for recommending rejection. While there are engineering efforts to aggregate and standardize existing benchmarks, the contribution does not extend much beyond that.

One could argue that such work helps facilitate the benchmarking process, and I appreciate the effort the authors put into building the software. However, I do not think this type of contribution is suitable for a venue like ICLR. I acknowledge that others might hold a different opinion, so I am setting my confidence level to 3.

**Strengths:**

1. There are interesting ideas in this work that could be expanded and delved into further. For example, the authors attempt to create a unified benchmark for DNA and RNA tasks, which raises the potentially interesting topic of studying the cross-modality transferability of GFMs. This aspect could be expanded and investigated more deeply to provide valuable insights.
2. Having a unified platform like this can enhance reproducibility in GFM development and evaluation.
3. The platform includes user-friendly features such as an API, an online hub, and a leaderboard.

**Weaknesses:**

1. As mentioned in the summary, this work generally lacks novelty.
2. One of the four main contributions claimed by the authors, Data scarcity and bias, is vaguely stated and not particularly valid. First, the discussion seems to conflate training data and evaluation data. This work addresses only the evaluation side, so the writing should focus on that. Second, since the data sources are four well-established benchmark datasets, there is no additional contribution to resolving the data scarcity issue, as no new dataset is introduced.
3. The authors did not discuss or justify the choice of the four benchmark datasets. There is no comprehensive review of existing benchmarks, which I think is conventional for such work. For example, the benchmarks from NT (https://www.biorxiv.org/content/10.1101/2023.01.11.523679v4), BEND (https://openreview.net/forum?id=uKB4cFNQFg), and Tang et al. (https://www.biorxiv.org/content/10.1101/2024.02.29.582810v2.full.pdf) are well-established and recognized benchmarks in the field but none were mentioned in the manuscript.
4. The manuscript contains many claims that lack supporting evidence and concrete examples. For instance:
a. The authors mention metric variability but do not show empirical evidence. How would using the original metrics from each benchmark affect the results?
b. The authors claim that this work helps test models on underrepresented sequences and tasks, but there is no further explanation or evidence on why or how this work addresses this issue.
5. The writing could be improved for conciseness and clarity. And there are some overly repetitive statements throughout the article.

**Questions:**

There are two repeated paragraphs in lines 154-167, as well as some other typos in the manuscript.

---

> ### Author Response · Authors · 2024-11-20
> **Clarifications and Rebuttal**
>
> **Weakness**
>
> **W1 Reply:** Our primary objective with this work is not to claim theoretical or technical novelty but to address the critical gap in the AI4Genomics field, i.e., a lack of universal genomic benchmarking tools. We focus on providing a comprehensive solution for both expert and novice genomics researchers, which includes support for diverse models and benchmark suites, as well as auxiliary features like AutoBench to streamline the evaluation process. We believe that such practical contributions can significantly benefit the community by enabling standardized, reproducible, and accessible genomic model evaluations.
>
> **W2 Reply:** We apologize for any misunderstanding regarding this point.
> - Our contributions focus exclusively on the benchmark (i.e., evaluation) phase, and we do not claim to directly resolve the issue of data scarcity in pretraining by introducing new datasets. Instead, our aim is to address the challenges of data scarcity and bias indirectly through tools like AutoBench, which enables fast and intuitive performance visualizations across diverse benchmarks.
> - The included benchmarks reflect potential biases that may arise during the model’s early training phases (e.g., proof-of-concept training). By leveraging AutoBench, developers can identify and mitigate these issues early in the development process. For instance, the tool allows developers to train and validate smaller model variants using the built-in benchmark suites, uncovering potential data scarcity or bias-related issues proactively. This diagnostic approach enhances the utility of existing datasets by promoting more informed and robust model development, even within the constraints of data availability.
>
> **W3 Reply:** Thank you for highlighting the importance of justifying our benchmark choices.
> - The principles guiding our selection are diversity and popularity. As described in the *Benchmark Suites* section, RGB serves as an RNA-oriented benchmark, PGB is tailored for plant DNA tasks, GUE is designed for multispecies (e.g., human, fungi, bacteria) model benchmarking, and RGB has been widely utilized in recent DNA GFM studies. These benchmarks were chosen to ensure broad coverage of data types and species, addressing a variety of genomic modeling scenarios.
> - We acknowledge the value of including additional well-established benchmarks, such as those from NT, BEND, and Tang et al. However, as these benchmarks are relatively new, we are in the process of incorporating them into our framework. This iterative inclusion is part of the long-term evolution of the project, ensuring it remains robust and comprehensive over time.
> - The primary contribution of this work lies in the universal benchmarking framework, *AutoBench*, which aims to standardize and streamline the evaluation process. While incorporating additional benchmarks is indeed meaningful, it represents incremental progress and does not diminish the validity or core contributions of *AutoBench*. We appreciate your suggestion and will prioritize integrating these benchmarks in future updates as the project evolves.
>
> **W4 Reply:** Thank you for pointing out the need for additional clarity and evidence. We apologize for the oversight and address your concerns below:
> - **Metric Variability:** Metric variability has been discussed in prior work (e.g., [1]) and stems from the limitations of certain metrics in specific contexts. For instance, the original performance metric for some benchmarks, such as accuracy, is less sensitive to class imbalance compared to metrics like F1-score. In PGB's splice site prediction task, the dataset is highly imbalanced, with over 98% of examples being negative. Using accuracy alone could lead to misleading conclusions about model performance, as it may not capture fairness or efficacy across classes. Our framework emphasizes metrics better suited to such scenarios, ensuring more reliable and fair evaluations.
> - **Testing on Underrepresented Sequences and Tasks:** The broad coverage of benchmarks in our framework involves evaluating models on millions of sequences across diverse species and sequence types. This diversity naturally exposes models to underrepresented sequences and tasks, such as those from less-studied organisms or rare genomic patterns. By including benchmarks like PGB and GUE, which encompass a wide variety of species and genomic contexts, our framework helps uncover limitations in model generalizability and performance across these underrepresented domains.
>
> We appreciate your feedback and will ensure these points are better articulated in the manuscript with supporting examples and evidence.
>
> [1] [Rethinking performance measures of rna secondary structure problems](https://arxiv.org/abs/2401.05351)
>
> **W5 Reply:** We appreciate your suggestions for presentation refinement and will address the problem in the revision.

---

### Official Review · Reviewer_sW8e · 2024-11-04

**Soundness:** 2
**Presentation:** 1
**Contribution:** 2
**Rating:** 3
**Confidence:** 3

**Summary:**

This paper introduces a meta-benchmark for transfer learning that integrates tasks from four existing benchmark sets. They revise metrics used in these benchmarks to account for class imbalance. The authors implemented their evaluation protocol in a Python package they claim is easily extendable and flag this as their primary contribution.

Orthogonally, the paper presents benchmarking results for several candidate foundation models for genomics. In addition to DNA language models, they include models trained on RNA. The authors highlight the strong performance of the RNA language model OmniGenome.

**Strengths:**

1. The authors address the problem of evaluating genomic foundation models, which is an important problem.
2. They appear to have developed a comprehensive, well-documented framework for benchmarking.
3. They convincingly demonstrated that current language models trained on DNA sequences do not learn representations of RNA sequences as effectively as those trained purely on RNA sequences.
4. They provide strong evidence that supervising models on annotations of RNA structure provides these models with important information.

**Weaknesses:**

1. The paper would be stronger if the evaluations were more comprehensive. The authors could have included newer benchmarks—like BEND (ICLR 2024) or those included in the recent preprint by Tang and Koo (https://doi.org/10.1101/2024.02.29.582810) —that address important limitations of the previous benchmarks.
2. It would also benefit from including additional language model baselines, like PlantCaduceus or GPN.
3. It would also be worthwhile to evaluate models other than language models, since these have been shown to outperform language models on transfer learning tasks.
4. I am not convinced of their claim that “OmniGenome generalizes well to DNA-based tasks, likely due to shared sequence motifs and structural similarities between RNA and DNA.” I do not think this explanation is true for noncoding elements. It is more likely that their benchmarking needs to be more thorough. From their comparisons, it is also unclear whether OmniGenome generalizes well—only that it generalizes comparably to language models trained on DNA.

**Questions:**

Does OmniGenome utilize structural information in its input?

---

> ### Author Response · Authors · 2024-11-20
> **Clarifications and Rebuttal**
>
> **Weakness**
>
> **W1 Reply:**  Thank you for highlighting the importance of including the BEND benchmark and the recent preprint by Tang and Koo. We recognize the value of wide coverage in benchmarking GFMs. As part of our ongoing efforts, we are actively adapting and integrating benchmarks, including those mentioned, and they will be incorporated in the forthcoming updates to our framework. It is important to note that the primary objective of our package is to incubate a universal genomic benchmarking and application toolbox, while the supported GFMs and benchmarks are add-on contributions in essence. On the one hand, it standardizes and automates benchmarking for RNA and DNA sequences, addressing reproducibility issues commonly encountered with genomic foundation models. On the other hand, it simplifies access to bioinformatics applications through an intuitive interface, as demonstrated in our tutorials (e.g., RNA design, RNA augmentation, and embedding extraction).
> Finally, we believe this incremental work does not represent a fundamental limitation of our current contributions. The four built-in benchmark suites already cover a wide range of use cases, including genomes (RNA and DNA) and species diversity (e.g., plants, humans, and bacteria). We appreciate your feedback and remain committed to refining and expanding the framework for broader applicability.
>
> *W2 Reply:* Thank you for pointing out the potential benefit of including additional model supports, i,e., PlantCaduceus and GPN. Expanding the coverage of recent GFMs in GFMBench is a priority for us, but the development timeline has exceeded our planned workloads, making it challenging to incorporate these models before the official release of the package. However, we are actively working on integrating these baselines as soon as long-term workloads. We appreciate your understanding and your constructive feedback, which will help us further enhance the utility of our framework.
>
> *W3 Reply:* Our current package is primarily designed to automate the evaluation of pre-trained GFMs, as we are focused on creating a standardized and streamlined benchmark pipeline for the emerging field of GFMs. This emphasis aligns with our goal of addressing the immediate need for benchmarking in this rapidly growing area. That said, we recognize the importance of evaluating other types of models, especially given their demonstrated strengths in transfer learning tasks. As we progress towards stable releases, we plan to expand our framework to include support for a broader range of models and will prioritize new features based on user feedback and community needs. We appreciate your input and look forward to enhancing the package to better serve diverse research goals.
>
> *W4 Reply:* Our statement is based on empirical observations derived from our experimental results. It is intended as a hypothesis, rather than a guranteed conclusion, to spark future research into why an RNA-based model like OmniGenome performs well on DNA tasks, even without specialized training. To the best of our knowledge, there is currently no specific benchmark designed to explain this phenomenon comprehensively.
> Regarding the concern about generalizability, we note that OmniGenome demonstrates necessary generalizability in the current benchmarking pipelines. For instance, it performs well across species (e.g., from plants to humans, as shown in Tables 3 and 4) and between modalities (e.g., from RNA to DNA, as seen in Table 1). We acknowledge that additional benchmarks and more thorough evaluations could provide further insights. We are committed to enhancing the benchmarking process in future iterations and look forward to refining our understanding as new evidence and benchmarks become available.
>
> *Questions*:
>
> *Q1 Reply:* Yes. According to the original publication, OmniGenome incorporated structural information in its modeling process. This integration of structural features is one of the key factors that may contribute to its generalizability, as it leverages potential similarities in structures and sequence motifs across RNA and DNA tasks.

---

> > ### Comment · Reviewer_sW8e · 2024-11-27
> >
> > In my review, I made suggestions to improve the model analysis included in the paper. Based on the authors' reply, I am under the impression that they view this analysis as a minor contribution next to their software package, and my suggestions as impertinent. I think the analysis is important precisely because it is what a powerful software package should enable in the first place. As it stands, I am unsure about the value of their contribution.
> >
> > Would it be fair to say that OmniGenome has an advantage over other models because it utilizes structural features in its input?

---

### Meta-Review · Area_Chair_Zd62 · 2024-12-17

**Metareview:**

This paper is about developing a better benchmark for transfer learning tasks. The construct a benchmark that is a composition of existing benchmarks sets and implement it in python. Reviewers noted the value of benchmarking models so that the community can assess performance and choose among models. There were concerns that some of the claims did not rest firmly on the evidence presented and that the novelty was not clearly described in relation to existing literature. The authors offered a rebuttal acknowledging that some of the concerns would be addressed in the subsequent research work. At this time the concerns about the claims and novelty place this work below the bar for acceptance.

**Additional Comments On Reviewer Discussion:**

The reviewers conducted a review of the paper, and the authors offered a point-by-point rebuttal. Reviewer sW8e made an important point in the reply to the author rebuttal that it speaks to what the overall benefit to the community would be for having read the paper.

---

### Decision · Program_Chairs · 2025-01-22

Reject